# Pupil diameter is not an accurate real-time readout of locus coeruleus activity

Marine Megemont[1†], Jim McBurney-Lin[1,2†], Hongdian Yang[1,2*]

[1]Department of Molecular, Cell and Systems Biology, University of California, Riverside, Riverside, United States; [2]Neuroscience Graduate Program, University of California, Riverside, Riverside, United States

**Abstract** Pupil diameter is often treated as a noninvasive readout of activity in the locus coeruleus (LC). However, how accurately it can be used to index LC activity is not known. To address this question, we established a graded relationship between pupil size changes and LC spiking activity in mice, where pupil dilation increased monotonically with the number of LC spikes. However, this relationship exists with substantial variability such that pupil diameter can only be used to accurately predict a small fraction of LC activity on a moment-by-moment basis. In addition, pupil exhibited large session-to-session fluctuations in response to identical optical stimulation in the LC. The variations in the pupil–LC relationship were strongly correlated with decision bias-related behavioral variables. Together, our data show that substantial variability exists in an overall graded relationship between pupil diameter and LC activity, and further suggest that the pupil–LC relationship is dynamically modulated by brain states, supporting and extending our previous findings (Yang et al., 2021).

## Editor's evaluation

This work is of interest to those studying neuromodulatory systems, such as the noradrenergic Locus Coeruleus (LC), because it extends prior studies on the relationship between neuronal activity and pupil diameter in awake and anaesthetized animals. Consistent with previous work, the authors show that there is a monotonic relationship between pupil diameter and number of LC spikes, however, they find that pupil diameter can only be used to predict a subset of LC spiking activity levels. This work supports the view that the relationship between pupil diameter and LC spiking may vary dynamically depending on behavioral state.

*For correspondence:
hongdian@ucr.edu

†These authors contributed equally to this work

Competing interest: The authors declare that no competing interests exist.

## Introduction

Fluctuations of brain states, such as arousal and attention, strongly impact sensory processing, decision-making, and animal behavior (*Harris and Thiele, 2011*; *Lee and Dan, 2012*; *Thiele and Bellgrove, 2018*; *Petersen, 2019*; *McCormick et al., 2020*). It is thus critical to understand the neural substrates of brain states and how state changes can account for the variability embedded in neuronal and behavioral data (*Cano et al., 2006*; *Poulet and Petersen, 2008*; *Polack et al., 2013*; *Fu et al., 2014*; *Zagha and McCormick, 2014*; *Lee et al., 2020*). Changes in pupil diameter under constant luminance are tightly linked to states of arousal and attention (*McGinley et al., 2015a*; *Joshi and Gold, 2020*). Dynamic pupil responses are associated with membrane potential fluctuations, sensory evoked responses, salience detection, error estimation, decision-making, and task performance (*Kahneman and Beatty, 1966*; *Bijleveld et al., 2009*; *Nassar et al., 2012*; *de Gee et al., 2014*; *de Gee et al., 2020*; *Reimer et al., 2014*; *Vinck et al., 2015*; *McGinley et al., 2015b*; *Lee and Margolis, 2016*; *Schriver et al., 2018*; *Schriver et al., 2020*; *Kucewicz et al., 2018*; *Ebitz and Moore, 2019*). As a result, pupil diameter has been widely used to monitor brain states and their neural substrates.

A multitude of neural circuits have been implicated in mediating brain state and pupil size changes, most notably the neuromodulatory systems (*Yu and Dayan, 2005*; *Lee and Dan, 2012*; *Thiele and Bellgrove, 2018*; *Joshi and Gold, 2020*). The locus coeruleus (LC)–noradrenergic system has long been thought to play a critical role in controlling arousal and attention (*Aston-Jones et al., 1999*; *Berridge and Waterhouse, 2003*; *Aston-Jones and Cohen, 2005*; *Sara, 2009*; *Sara and Bouret, 2012*), and LC activity closely tracks brain states and cognitive processes (*Foote et al., 1980*; *Aston-Jones and Bloom, 1981*; *Berridge and Foote, 1991*; *Aston-Jones et al., 1994*; *Rajkowski et al., 1994*; *Usher et al., 1999*; *Takahashi et al., 2010*; *Carter et al., 2010*; *Eschenko et al., 2012*; *Vazey and Aston-Jones, 2014*; *Kalwani et al., 2014*; *Varazzani et al., 2015*; *Bouret and Richmond, 2015*; *Fazlali et al., 2016*; *Swift et al., 2018*). Importantly, work mainly in the past decade has provided correlative and causal evidence linking pupil size changes to LC activity (*Rajkowski et al., 1994*; *Murphy et al., 2014*; *Varazzani et al., 2015*; *Joshi et al., 2016*; *Reimer et al., 2016*; *de Gee et al., 2017*; *Liu et al., 2017*; *Breton-Provencher and Sur, 2019*; *Hayat et al., 2020*; *Privitera et al., 2020*), leading to the increased utilization of pupil diameter as a noninvasive readout of LC (*Aston-Jones and Cohen, 2005*; *Gilzenrat et al., 2010*; *Preuschoff et al., 2011*; *Konishi et al., 2017*; *Gelbard-Sagiv et al., 2018*; *Zhao et al., 2019*; *Aminihajibashi et al., 2020*; *Clewett et al., 2020*). However, a few recent studies demonstrated that the correlation between pupil and LC could be neuron- and task epoch-specific (*Joshi et al., 2016*; *Breton-Provencher and Sur, 2019*; *Yang et al., 2021*), raising the possibility that pupil diameter can be dissociated from LC activity. To the best of our knowledge, we do not know to what extent pupil diameter is linked to LC activity. More importantly, we do not know whether and how pupil diameter can be used to make accurate inferences of LC activity on a moment-by-moment basis.

To address these questions, we recorded spiking activity from optogenetically tagged LC neurons simultaneously with pupil diameter in head-fixed mice trained to perform a tactile detection task (*McBurney-Lin et al., 2020*; *Yang et al., 2021*). We established a graded relationship between pupil and LC, where pupil dilation increased monotonically with LC spiking activity. However, this relationship exists with substantial variability such that pupil size changes can only accurately predict a small fraction of LC spiking on a moment-by-moment basis. Using optogenetics to activate LC neurons, we showed that pupil responses exhibited large session-to-session fluctuations to identical optical stimulation, despite stable LC responses. Notably, decision bias-related behavioral variables explained the variations in the pupil–LC relationship. Together, our data show that substantial variability exists in an overall positive relationship between pupil diameter and LC activity, and that only under limited conditions can pupil be used as an accurate real-time readout of LC. Our work further suggests that brain states dynamically modulate the coupling between pupil and LC.

## Results

We recorded spiking activity from optogenetically tagged single units in the LC together with pupil diameter in head-fixed mice during behavior (*Figure 1a*). To quantify a graded relationship between pupil size changes and LC spiking, we first grouped adjacent spikes into individual clusters (*Hahn et al., 2010*; *Yu et al., 2017*) based on each unit's median interspike interval (*Figure 1b*, *Figure 1—figure supplement 1*, Methods). The magnitude of pupil responses following a spike cluster (quantified in a 6-s window from cluster onset) progressively increased with cluster size (the number of spikes in a cluster, *Figure 1c, d*). The latency of peak pupil diameter did not systematically vary with cluster size and ranged between 2.5 and 4 s (*Figure 1—figure supplement 2*). This latency is consistent with our previous report (*Yang et al., 2021*). Overall, we found a positive, monotonic relationship between peak pupil diameter and LC cluster size in the majority of paired recordings (linear regression $R^2 >$ 0.6 in 13 out of 19 paired recordings, *Figure 1e*), in line with previous findings in nonhuman primates (*Varazzani et al., 2015*; *Joshi et al., 2016*). Similar relationships held when pupil responses were quantified as % changes from baseline (*Cazettes et al., 2021*) or the time derivatives of pupil (*Reimer et al., 2016*; *Yang et al., 2021*; *Figure 1—figure supplement 3*). However, substantial variations existed in the relationship (linear slopes ranging from 0.12 to 0.51. 0.24 ± 0.11, mean ± standard deviation [SD], $n$ = 13), indicating variable couplings between pupil and LC neurons.

Although pupil diameter exhibited an overall monotonic relationship with LC spiking, it did not necessarily warrant pupil diameter being an accurate readout of LC activity. We tested the extent to which pupil size changes can be used as a proxy for LC activity, that is, can we use pupil diameter

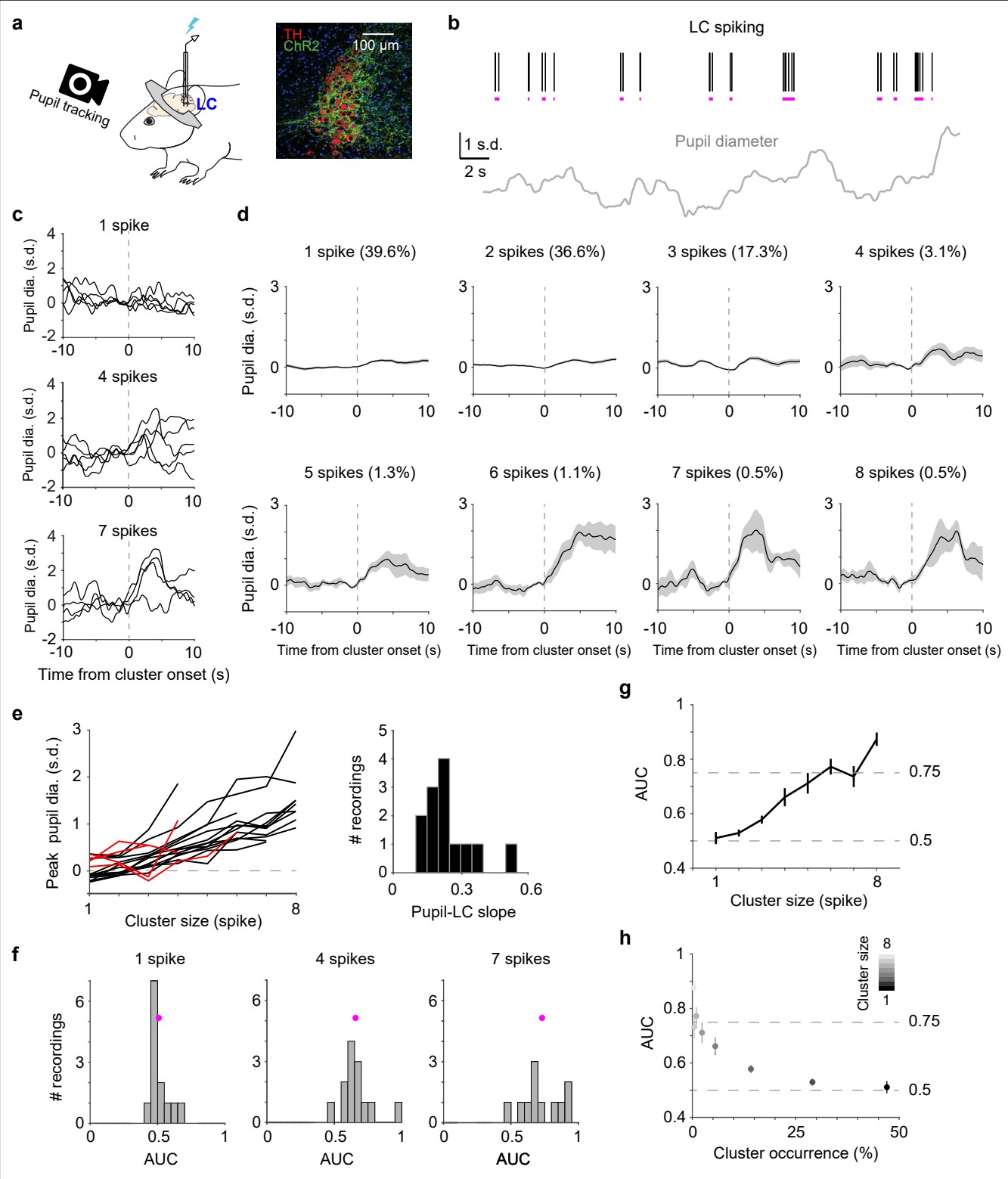

**Figure 1.** Correlating locus coeruleus (LC) activity to pupil responses. (**a**) Left: schematic of experimental setup for simultaneous pupil and LC recording/ optical stimulation in head-fixed mice. Lightning bolt: light pulse. Right: expression of ChR2 in a DBH;Ai32 mouse (dopamine beta hydroxylase, DBH; ChR2-EYFP: green; tyrosine hydroxylase, TH: red). (**b**) Example simultaneously recorded LC spike raster and *z*-scored pupil diameter. Vertical black lines represent individual spikes. Horizontal magenta lines indicate spike clusters. (**c**) Example LC spike cluster-triggered pupil responses for cluster sizes 1, 4, and 7. (**d**) Mean LC cluster-triggered pupil responses (± standard error of the mean [SEM]) for cluster sizes 1 through 8 with occurrence (%) indicated

*Figure 1 continued on next page*

*Figure 1 continued*

in an example recording. (**e**) Left: the relationship between peak pupil diameter and LC cluster size for each paired recording. Curves with linear regression $R^2 > 0.6$ are in black ($n = 13$), <0.6 in red ($n = 4$). Two recordings with limited cluster sizes (<3) were not suitable for linear regression and not included here. Right: histogram of the linear slopes for curves with $R^2 > 0.6$. For f–h, the 13 recordings with $R^2 > 0.6$ were included. (**f**) Histograms of area under the curve (AUC) values when using peak pupil diameter to predict the associated cluster sizes 1, 4, and 7. Magenta dot: mean. (**g**) Group mean AUC values when using peak pupil diameter to predict the associated cluster sizes 1 through 8. (**h**) Replot of (**g**) by showing the occurrence (abscissa) associated with each cluster size (gray scale).

The online version of this article includes the following source data and figure supplement(s) for figure 1:

**Source data 1.** Source data for *Figure 1*.

**Figure supplement 1.** Locus coeruleus (LC) and pupil recordings in mice.

**Figure supplement 2.** The relationship between the latency of peak pupil diameter and locus coeruleus (LC) spike cluster size.

**Figure supplement 3.** The relationship between pupil size changes and locus coeruleus (LC) spike cluster.

**Figure supplement 4.** Group mean area under the curve (AUC) values when using peak pupil diameter to predict the associated cluster sizes 1 through 8 from all recordings ($n = 19$).

**Figure supplement 5.** Group mean probability distribution of locus coeruleus (LC) spike clusters ($n = 19$).

to make accurate inferences of LC spiking on a moment-by-moment basis? We asked how well an ideal observer (*Green and Swets, 1966*) can predict LC cluster size given the associated peak pupil responses (Methods). Receiver-operating-characteristic (ROC) analysis showed that as cluster size increased, peak pupil diameter can better predict LC activity (*Figure 1f, g*, *Figure 1—figure supplement 4*). However, only peak pupil diameter associated with large clusters (≥5–6 spikes) can achieve a performance threshold of $d' = 1$ (translates to ~0.75 area under the curve *Simpson and Fitter, 1973*, *Figure 1g*). Since larger clusters occurred less frequently (*Figure 1d, h*, *Figure 1—figure supplement 5*), our data suggest that pupil dilation cannot accurately represent the majority of LC spiking activity but can serve as a good proxy for the infrequent (<10%) and strong LC activity in real time (*Figure 1h*).

Perhaps what is more interesting (and useful) is to assess whether we can directly use pupil diameter to infer the 'ground truth' – LC activity, without recording from LC. To do so, we first detected pupil dilation events based on zero-crossings of pupil derivatives (*Joshi et al., 2016*; *Figure 2a*) and quantified LC spike counts immediately preceding each dilation event (Methods). Compared with the analyses in *Figure 1*, this method did not require prior knowledge of LC activity for identifying pupil responses and yielded a similar pupil–LC relationship (*Figure 2—figure supplement 1*). Overall, LC spike counts were monotonically associated with pupil dilation amplitudes (*Figure 2b–d*). However, a wide range of spike counts preceded pupil events of similar sizes (*Figure 2b, c*). We asked how well an ideal observer can predict pupil dilation events given the associated LC activity and found that as pupil dilations became larger, LC spike counts could make better predictions on a moment-by-moment basis. However, only LC activity preceding the infrequent (<10%), large dilation events (>1.5–2 SD, *Figure 2—figure supplement 2*) performed beyond 75% threshold (*Figure 2e*). Finally, we tested how well we can use the detected pupil dilation events to predict LC activity. Similar to the previous results (*Figure 1g*), we found that only large pupil events can achieve good predictions (*Figure 2f*). Taken together, our data show that pupil diameter and LC spiking are well correlated in a graded manner and that the infrequent (<10%) but strong (>1.5–2 SD) pupil dilation events can be used to accurately and reliably predict LC activity in real time.

Our data presented so far were based on paired pupil–LC recordings, each consisting of a single opto-tagged LC unit. Next, we sought to test whether pupil size changes better reflect population-level LC activity instead of single neurons. To this end, we optogenetically activated groups of LC neurons and quantified the evoked pupil responses. Based on the stimulation parameters, we estimated an excitable volume on the order of 0.05–0.1 mm³, containing hundreds of LC neurons (*Figure 3a, b*, *Figure 3—figure supplement 1*, Methods). In a subset of experiments, the putatively same LC units were tracked (typically 1–5 days), based on opto-tagging, spike clustering, and waveform comparison (*Figure 3c, d*). Waveforms from the putatively same units were more similar than the waveforms from the putatively different units (*Figure 3e–g*). These putatively same units responded similarly to optical stimulation in different sessions (*Figure 3h*), suggesting a consistent transduction of optical stimulation to LC spiking activity. In contrast to stable LC responses, the same pupil exhibited variable dilations to optical stimulation under awake, nontask performing conditions

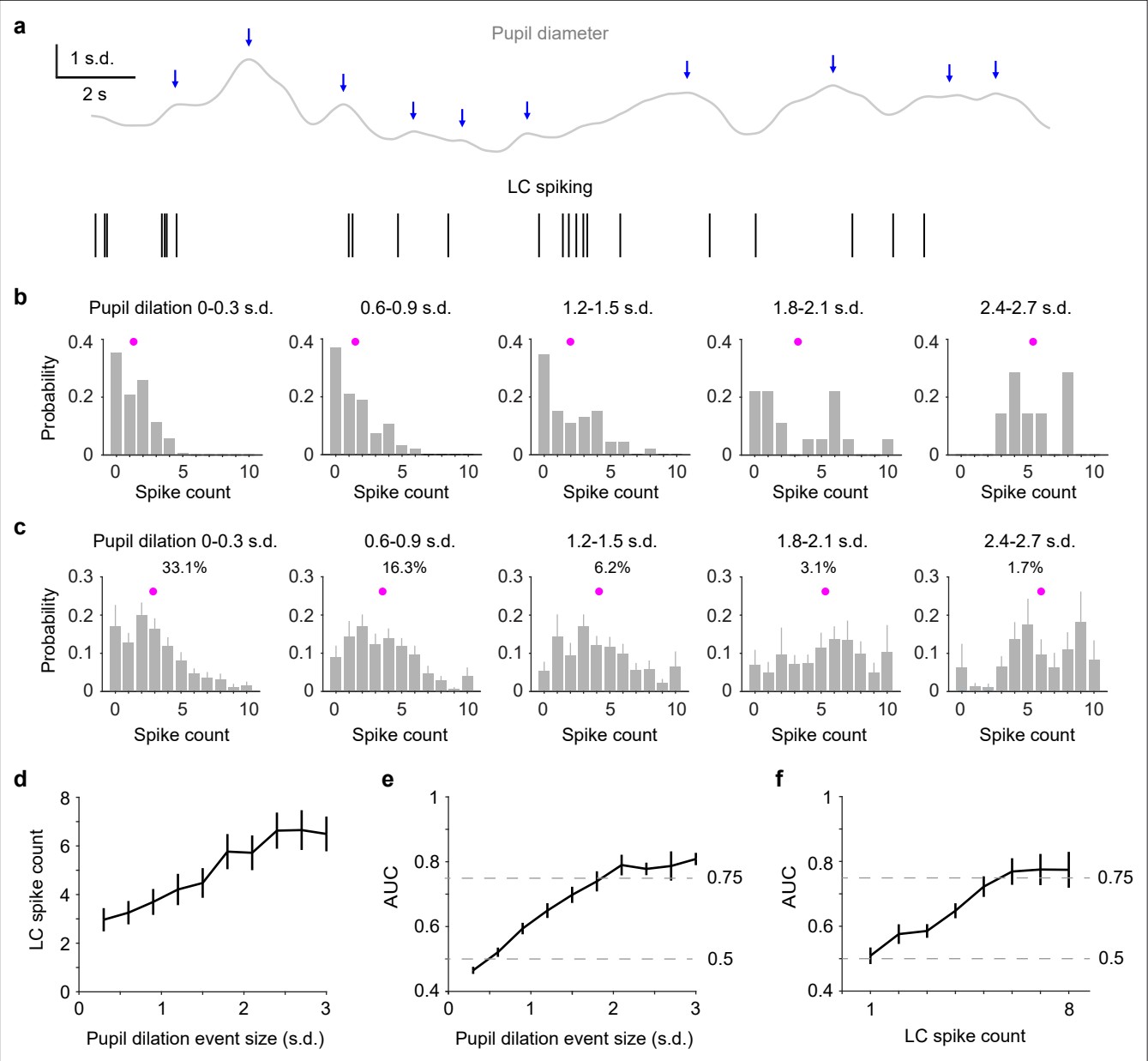

**Figure 2.** Reverse correlating pupil responses to locus coeruleus (LC) activity. (**a**) Example pupil–LC traces showing the detected pupil dilation events (blue arrows) based on zero-crossing of pupil derivatives. (**b**) Probability distributions of LC spike counts associated with pupil dilation events of similar sizes in an example recording. Magenta dot: mean. Pupil dilation events were binned every 0.3 standard deviation (SD). (**c**) Group mean probability distributions of LC spikes associated with pupil dilation events of similar sizes. Mean occurrences (%) of pupil dilation events were indicated. (**d**) Group mean relationship between LC spike counts and pupil dilation events binned every 0.3 SD from 0 to 3 SD. (**e**) Group mean area under the curve (AUC) values when using LC spike counts to predict the associated pupil dilation events binned every 0.3 SD from 0 to 3 SD. (**f**) Group mean AUC values when using the detected pupil dilation events to predict the associated LC spike counts 1 through 8, similar to *Figure 1g*.

The online version of this article includes the following source data and figure supplement(s) for figure 2:

**Source data 1.** Source data for *Figure 2*.

**Figure supplement 1.** Group mean relationship between peak pupil diameter and locus coeruleus (LC) spike counts using two different methods.

**Figure supplement 2.** Group mean probability distribution of the detected pupil dilation events (*n* = 19).

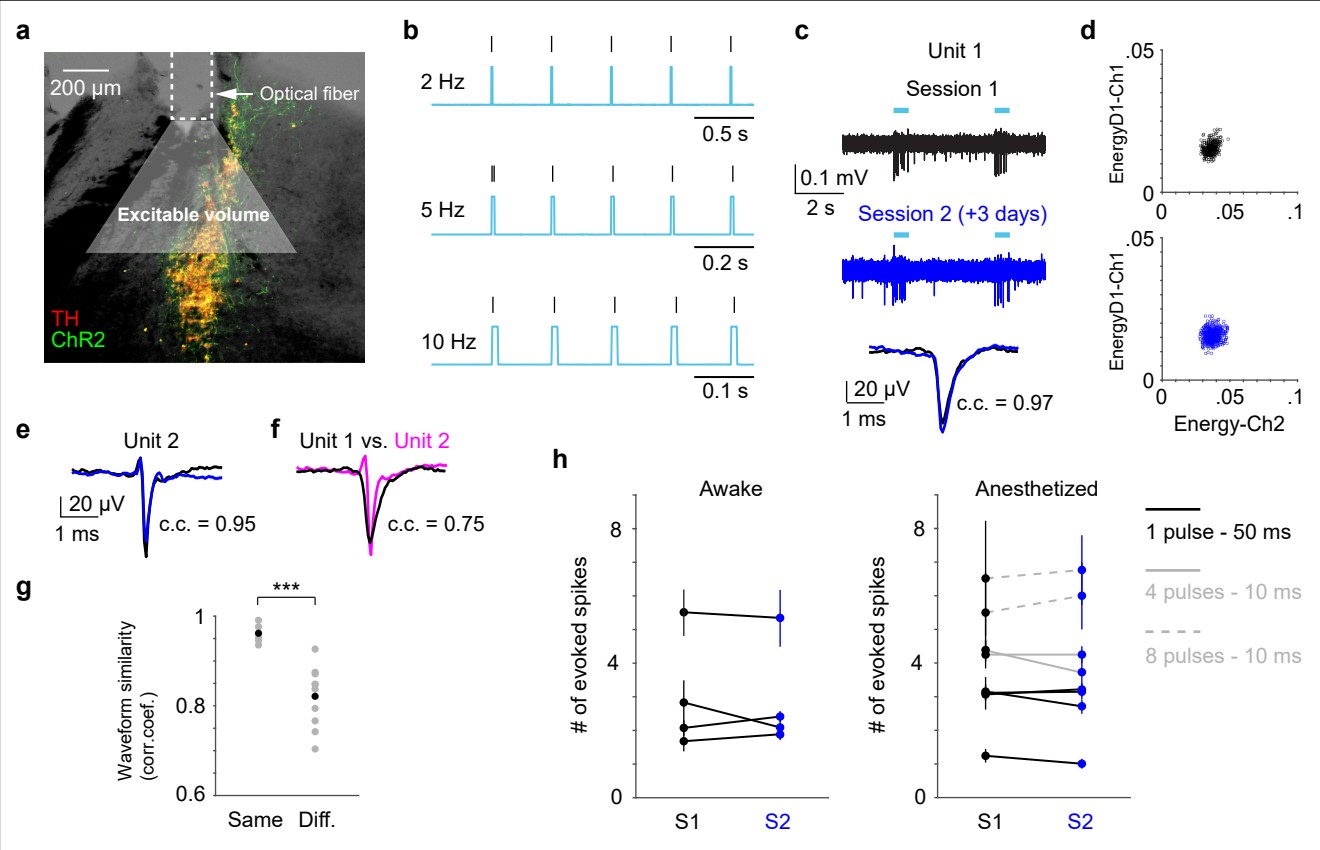

**Figure 3.** Locus coeruleus (LC) responses to optogenetic stimulation. (**a**) Example LC histological section illustrating optical fiber implant and the estimated excitable volume (light gray cone). Estimates were based on 10-mW laser power, 2.5 mW/mm² excitation threshold, 1.4 refractive index, and a 30° cylindrical cone. (**b**) Example spiking activity (vertical lines) from an opto-tagged LC unit in response to 10-ms blue pulse trains at different frequencies. (**c**) Example traces (top, middle) and waveforms (bottom) from a putatively same LC unit in response to optical stimulation (cyan bars) in two different sessions (3 days apart). Black and blue indicate an earlier and a later session (sessions 1 and 2), respectively. Waveforms from the two sessions were highly similar with Pearson correlation coefficient (c.c.) = 0.97. (**d**) Spike sorting diagrams corresponding to the two sessions shown in (**c**). The unit was identified in Ch1. (**e**) Waveforms from another putatively same unit in two sessions (1 day apart, waveform c.c. = 0.95). (**f**) Waveforms from the 2 units shown in (**c**) and (**e**) were less similar (session 1 unit 1 vs. session 1 unit 2, c.c. = 0.75). (**g**) Among the tracked 5 units, waveforms from the putatively same units in sessions 1 and 2 (Same) were more similar than waveforms from the putatively different units in session 1 (Different. Same vs. Different, Pearson correlation coefficient (c.c.), 0.96 ± 0.02 vs. 0.82 ± 0.07, mean ± standard deviation (SD), p = 6.6e−4, two-tailed rank sum test). Gray dots: individual pair. Black dots: group mean. (**h**) Responses from the putatively same units to optical stimulation (S1 vs. S2) during awake, nontask performing (4 units, left) and anesthetized (5 units, right) conditions. p > 0.05 for each S1 vs. S2 comparison, permutation test. Evoked spike counts were quantified in response to (1) single 50 ms pulse (solid black line, 4 units); or (2) four 10 ms pulses at 10 Hz (solid gray line, 2 units); or (3) eight 10 ms pulses at 5 Hz (dashed gray line, 2 units).

The online version of this article includes the following source data and figure supplement(s) for figure 3:

**Source data 1.** Source data for *Figure 3*.

**Figure supplement 1.** Locus coeruleus (LC) response to optogenetic stimulation.

(***Figure 4a, b***). Importantly, baseline pupil diameters were similar (0.71 vs. 0.75 mm, ***Figure 4—figure supplement 1***) and thus cannot explain the differences in evoked pupil responses. Group data from multiple mice further demonstrated that significant session-to-session fluctuations of pupil responses were prevalent but not directional (solid lines in ***Figure 4c, d***), that is, pupil responses in an earlier session (session 1) were not consistently higher or lower than in a later session (session 2). Therefore, such session-to-session fluctuations were not observable from group comparisons (***Figure 4—figure supplement 2***, ***Privitera et al., 2020***). To further test whether the variable pupil responses were due to (1) weak LC stimulation with 10 ms pulses, or (2) strong spontaneous pupil fluctuations during wakefulness, we performed the following experiments. First, we evoked pupil responses with stronger stimulation (50 ms pulses instead of 10 ms) in the awake condition. While baseline pupil diameters

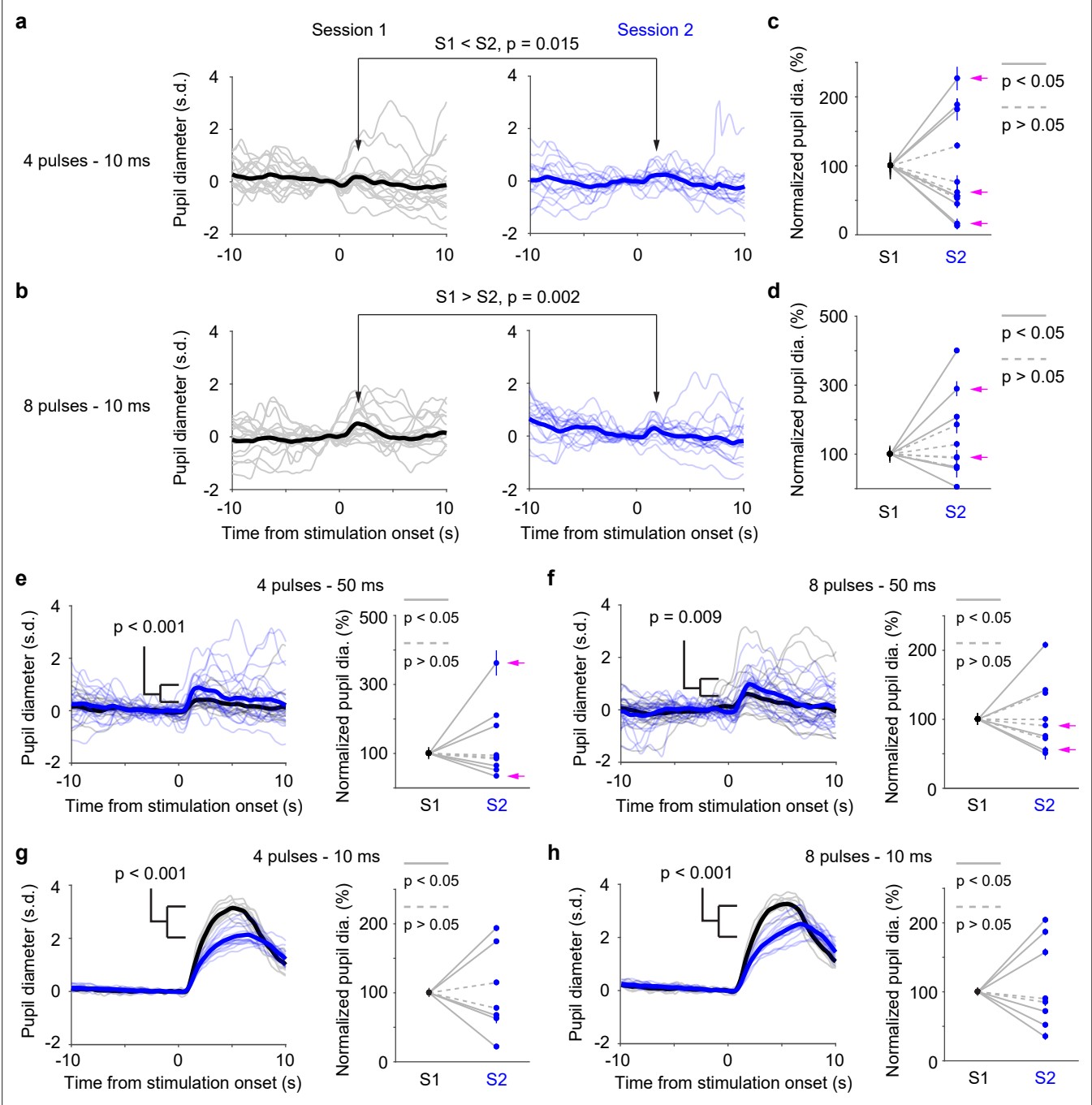

**Figure 4.** Pupil responses to locus coeruleus (LC) optogenetic stimulation. (**a**) Example responses from the same pupil to LC stimulation in two awake, baseline pupil-matched sessions (left and right) aligned to the onset of optical stimulation of four 10 ms pulses at 10 Hz. Thin curves: individual responses; thick curves: mean. Baseline pupil diameter S1 vs. S2, 0.71 vs. 0.75 mm. p values were based on permutation test. (**b**) Same as in (**a**), except that optical stimulation was eight 10 ms pulses at 10 Hz. (**a, b**) were from the same recording. (**c**) Group data showing pupil responses to optical stimulation of four 10 ms pulses at 10 Hz in awake, baseline pupil-matched sessions (12 paired sessions from 6 mice). To aid visualization, pupil responses in session 2 were normalized to session 1. Unnormalized data in *Figure 4—figure supplement 2*. Dots: mean peak pupil responses. Vertical lines: 95% confidence interval. Solid lines indicate significant difference (p < 0.05, permutation test). Session 1 always preceded session 2. Magenta arrows indicate same-day comparison. (**d**) Group data showing pupil responses to optical stimulation of eight 10 ms pulses at 10 Hz in awake, baseline pupil-matched sessions (11 paired sessions from 7 mice). Unnormalized data in *Figure 4—figure supplement 2*. Conventions are as in (**c**). (**e, f**) Left: example pupil responses from one recording. Conventions are as in (**a, b**), except that optical stimulations consisted of 50 ms pulses instead of 10 ms, and that pupil responses from the two sessions were overlaid. Baseline pupil diameter S1 vs. S2, 0.83 vs. 0.80 mm. Right: group pupil responses as in (**c, d**), except that optical stimulations consisted of 50 ms pulses instead of 10 ms. 9 paired sessions from 7 mice in (**e**), and 9 paired sessions from 7 mice in

*Figure 4 continued on next page*

*Figure 4 continued*

(**f**). Magenta arrows indicate same-day comparison. (**g, h**) Left: example pupil responses from one recording. Conventions are as in (**a, b**), except that the mouse was under anesthesia (2% isoflurane), and that pupil responses from the two sessions were overlaid. Baseline pupil diameter S1 vs. S2, 0.31 vs. 0.35 mm. Right: group pupil responses as in (**c, d**), except that mice were under anesthesia. 7 paired sessions from 3 mice in (**g**), and 8 paired sessions from 3 mice in (**f**).

The online version of this article includes the following source data and figure supplement(s) for figure 4:

**Source data 1.** Source data for *Figure 4*.

**Figure supplement 1.** Raw pupil traces for the two sessions used in *Figure 4a, b*.

**Figure supplement 2.** Unnormalized group pupil responses as shown in *Figure 4c, d*.

**Figure supplement 3.** The variability of pupil responses to locus coeruleus (LC) optical stimulation within individual sessions (Within) was comparable to that of across sessions (Across) in awake mice.

**Figure supplement 4.** Spontaneous pupil fluctuation was reduced during anesthesia.

**Figure supplement 5.** Simultaneous locus coeruleus (LC) and pupil responses to optical stimulation.

**Figure supplement 6.** List of all locus coeruleus (LC) recordings.

were similar between sessions, evoked pupil responses still fluctuated significantly (*Figure 4e, f*). In a subset of experiments, pupil exhibited substantial fluctuations in two sessions just several hours apart (4–6 hr, magenta arrows in *Figure 4c–f*). Further analysis showed that across-session variability of pupil responses was comparable to within-session variability (*Figure 4—figure supplement 3*, Methods). In addition, for the paired sessions that exhibited significantly different responses to optical stimulation (solid lines in *Figure 4c–f*), only a small subset exhibited larger across-session variability than within-session variability (2 pairs out of 12 under 10 ms condition, and 3 pairs out of 11 under 50 ms condition, Methods). Next, we stimulated LC with 10 ms pulses under anesthesia (2% isoflurane) to minimize spontaneous pupil fluctuations (*Figure 4—figure supplement 4*). Evoked pupil responses were noticeably larger compared with the awake condition in the example recordings, possibly due to a more constricted baseline pupil size under anesthesia (*Figure 4g, h*, left vs. *Figure 4a, b*, 0.3 vs. 0.7 mm). Nevertheless, pupil responses to optical stimulation exhibited substantial session-to-session fluctuations (*Figure 4g, h*). Additional examples of a simultaneously recorded LC unit and pupil diameter in responses to optical stimulation are in *Figure 4—figure supplement 5*. In summary, pupil responses showed large session-to-session fluctuations to identical LC stimulation.

What may underlie the variable pupil responses? We found that the variations in the relationship between peak pupil diameter and LC cluster size (as in *Figure 1e*) were strongly correlated with hit rate and decision bias during task performance (*Figure 5a*). This effect was not likely due to linear fitting of nonlinear relationships (all linear fits are of $R^2 > 0.85$. $0.92 \pm 0.05$, mean ± SD, $n = 9$), and the results held when the analysis of pupil–LC relationship was restricted to nonlicking periods only (*Figure 5b*, Methods). Therefore, although mice licked more during sessions of higher hit rate and

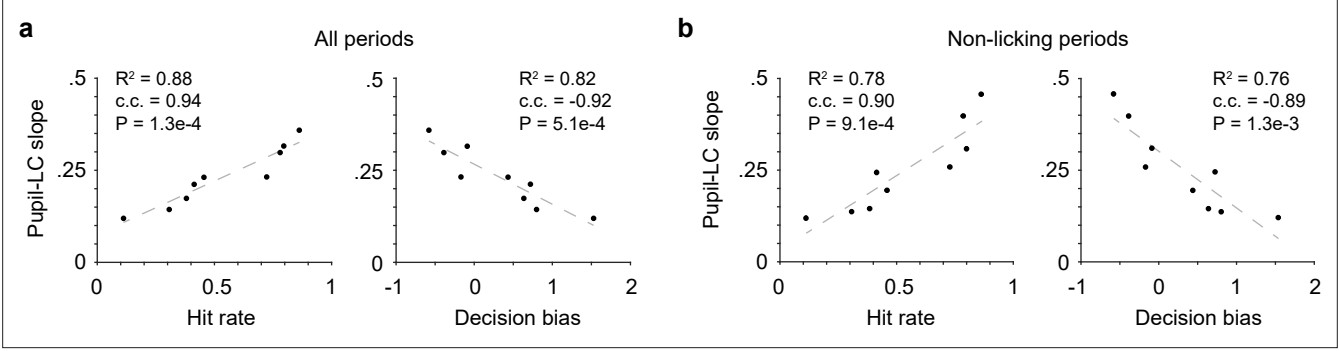

**Figure 5.** Pupil–locus coeruleus (LC) coupling correlated with decision-bias-related variables. (**a**) The variations in the relationship between peak pupil diameter and LC cluster size (linear slopes in *Figure 1e*) were strongly correlated with Hit rate (left) and decision bias (right, $n = 9$). c.c., Pearson correlation coefficient. (**b**) The relationships in (**a**) held when pupil–LC slopes were quantified in nonlicking periods only.

The online version of this article includes the following source data for figure 5:

**Source data 1.** Source data for *Figure 5*.

lower decision bias, the results cannot be fully explained by a potentially stronger pupil–LC coupling during licking periods. Based on these findings, we conclude that decision bias-related behavioral variables could explain, at least in part, the variations in the pupil–LC relationship. Since fluctuations of these behavioral variables reflect state changes such as impulsivity, motivation, and task engagement (*Dickinson and Balleine, 1994*; *Mayrhofer et al., 2013*; *Berditchevskaia et al., 2016*; *Allen et al., 2019*; *McBurney-Lin et al., 2020*), our results suggest that the coupling between pupil and LC is state dependent.

## Discussion

In the current study, we have shown that pupil diameter has an overall positive and monotonic relationship with LC spiking activity. However, substantial variability exists in this relationship that only the infrequent and large pupil dilation events (>1.5–2 SD amplitude, <10% occurrence) can accurately predict LC spiking on a moment-by-moment basis. In addition, pupil responses exhibit large session-to-session fluctuations to identical optical stimulation in the LC. Decision bias-related behavioral variables could explain the variations in the pupil–LC relationship. Together, our results strongly caution treating pupil dilation as a real-time readout of LC activity. Averaging multiple repeats/trials of similar pupil responses would yield a much more accurate prediction of LC activity.

We used two methods to establish the pupil–LC relationship: detecting LC activity then linking to the following pupil responses (*Figure 1*); and detecting pupil dilation then linking to the preceding LC activity (*Figure 2*). Both methods yielded similar pupil–LC relationships with the conclusion that only the infrequent, large pupil responses can accurately predict LC spiking on a moment-by-moment basis. Large pupil or LC responses have been reported to correlate with a variety of task-related processes, including sensory cue, decision formation, positive feedback, choice bias, and action (*Rajkowski et al., 1994*; *Usher et al., 1999*; *Kalwani et al., 2014*; *Bouret and Richmond, 2015*; *de Gee et al., 2017*; *de Gee et al., 2020*; *Schriver et al., 2020*; *Yang et al., 2021*). In light of this, our work suggests that the infrequent but strong pupil dilation events can be used as an accurate inference of LC activation in response to sensory stimuli and decision-making processes. However, as discussed below, in general pupil and LC likely respond to task-related processes differently, leading to variations in their relationship.

Recent evidence has uncovered considerable heterogeneity within the LC nucleus, including molecular compositions, physiological properties, released transmitters, and projection targets (*Robertson et al., 2013*; *Chandler et al., 2014*; *Chandler et al., 2019*; *Schwarz and Luo, 2015*; *Schwarz et al., 2015*; *Kempadoo et al., 2016*; *Hirschberg et al., 2017*; *Uematsu et al., 2017*; *Totah et al., 2018*; *Borodovitsyna et al., 2020*). Our data support these findings (*Figure 4—figure supplement 6*). Therefore, it is possible that pupil diameter is dynamically coupled with different LC subgroups that are differentially engaged during cognitive processes. However, this is insufficient to explain the session-to-session fluctuations of pupil responses to LC stimulation, since we likely activated a heterogenous group of LC neurons that exhibited similar session-to-session responses to optical stimulation.

The fact that the putatively same neurons tracked across days exhibited similar responses to optical stimulation cannot fully establish the long-term stability of population LC response because slow changes in the tissue due to tetrode/optical fiber implant (gliosis, inflammation, etc.) could alter light transmission to the neurons that were not recorded. However, several lines of evidence in our study did not favor this possibility: (1) Pupil responses in a later session did not systematically or progressively differ from an earlier session (e.g., consistently larger or smaller, *Figure 4—figure supplement 2*); (2) Significant pupil response variability can be observed in sessions that were a few hours apart (*Figure 4*); (3) Across-session variability of pupil responses was largely comparable to within-session variability (*Figure 4—figure supplement 3*). However, optogenetic stimulation tends to synchronize neuronal activity, which may not reflect the physiological condition (*Totah et al., 2018*). Future experiments with the ability to record from multiple opto-tagged LC neurons simultaneously will further investigate the relationship between pupil diameter and population-level LC activity.

During wakefulness, the state of the brain is constantly fluctuating, both in the presence and absence of external stimuli (*Kenet et al., 2003*; *Fox et al., 2006*; *Fox and Raichle, 2007*; *Sakata and Harris, 2009*; *Luczak et al., 2009*; *Berkes et al., 2011*; *Harris and Thiele, 2011*; *Mohajerani et al., 2013*; *Romano et al., 2015*; *Petersen, 2019*; *McCormick et al., 2020*). Pupil response profiles can reflect different behavioral processes (*Schriver et al., 2020*), and pupil responses also can be

dissociated from cognitive processes (*Podvalny et al., 2019*). Our data extend these observations, supporting that LC and pupil respond to behavioral and cognitive variables differently (*Yang et al., 2021*).

Fluctuations of hit rate and decision bias reflect state changes such as impulsivity, motivation, and task engagement (*Dickinson and Balleine, 1994*; *Mayrhofer et al., 2013*; *Berditchevskaia et al., 2016*). Although mice licked more during high motivation or high engagement trials (*Berditchevskaia et al., 2016*; *Allen et al., 2019*; *McBurney-Lin et al., 2020*), our data show that licking alone cannot account for the tight correlation between the variations of the behavioral variables and the variations in the pupil–LC relationship (*Figure 5*), suggesting that pupil–LC coupling is brain state dependent.

How may brain states modulate pupil–LC coupling? Pupil size changes have been linked to activity in other brain areas and neuromodulatory systems, including the medial prefrontal cortex, the inferior colliculus, and cholinergic signaling (*Joshi et al., 2016*; *Reimer et al., 2016*; *Okun et al., 2019*; *Kucyi and Parvizi, 2020*; *Pais-Roldán et al., 2020*; *Sobczak et al., 2021*). A recent study found that pupil responses to dorsal raphe stimulation exhibited task uncertainty-dependent variations (*Cazettes et al., 2021*). Therefore, it is possible that in high motivation/engagement states, multiple circuits including the LC synergistically influence pupil size changes, resulting in the apparently stronger pupil–LC coupling. Future experiments are needed to elucidate how pupil and LC interact with these brain circuits during different behavioral contexts and cognitive processes.

Another possibility is that higher engagement states may be intimately associated with more 'uninstructed' movements as revealed by recent work (*Musall et al., 2019*), which can drive robust neuronal activity throughout the brain (*Musall et al., 2019*; *Steinmetz et al., 2019*; *Stringer et al., 2019*; *Salkoff et al., 2020*). Future studies with comprehensive movement monitoring will determine whether more frequent movements, both task-related and task-unrelated, during periods of high motivation/engagement underlie the stronger pupil–LC coupling.

# Materials and methods

**Key resources table**

| Reagent type (species) or resource | Designation | Source or reference | Identifiers | Additional information |
|---|---|---|---|---|
| Strain, strain background (*M. musculus*) | DBH-Cre | MMRRC | RRID:MMRRC_036778-UCD | |
| Strain, strain background (*M. musculus*) | Ai32 | JAX | RRID:IMSR_JAX:012569 | |
| Software, algorithm | BControl | Princeton University | https://brodylabwiki.princeton.edu/bcontrol | |
| Software, algorithm | WaveSurfer | HHMI Janelia | http://wavesurfer.janelia.org/ | |
| Software, algorithm | Matlab | MathWorks | RRID:SCR_001622 | |
| Software, algorithm | Janelia eye tracker | HHMI Janelia | N/A | |
| Software, algorithm | StreamPix | Norpix | RRID:SCR_015773 | |
| Software, algorithm | Illustrator | Adobe | RRID:SCR_010279 | |
| Other | Camera | PhotonFocus | DR1-D1312-200-G2-8 | |
| Other | Telecentric lens | Edmund Optics | 55–349 | |
| Other | Tetrode drive | *Cohen et al., 2012* | N/A | |
| Antibody | Anti-TH primary antibody | Thermo Fisher | OPA104050 RRID:AB_325653 | 1:1000 |
| Antibody | Secondary antibody | Thermo Fisher | A11008 RRID:AB_2534079 | 1:500 |

All procedures were performed in accordance with protocols approved by UC Riverside Animal Care and Use Committee. Mice were DBH-Cre (B6.FVB(Cg)-Tg(Dbh-cre) KH212Gsat/Mmucd, 036778-UCD, MMRRC); Ai32 (RCL-ChR2(H134R)/EYFP, 012569, JAX), or DBH-Cre injected with AAV5-EF1α-DIO-hChR2(H134R)-EYFP (UNC Vector Core), singly housed in a vivarium with reverse light–dark cycle (12 hr each phase). Male and female mice of 8–12 weeks were implanted with titanium head posts

as described previously (*Yang et al., 2016*). Procedures for microdrive construction and LC recording have been described previously (*Yang et al., 2021*). Briefly, custom microdrives with eight tetrodes and an optic fiber (0.39 NA, 200 µm core) were built to make extracellular recordings from LC neurons. Microdrive was implanted in the left LC. LC neurons were identified by optogenetic tagging of DBH+ neurons expressing ChR2, tail pinch response, and post hoc electrolytic lesions (*Yang et al., 2021*). For *Figures 1 and 2*, 19 single unit recordings (cluster quality measure $L_{ratio}$: 0.01 ± 0.005; firing rate: 1.65 ± 0.25 spikes/s; percent ISI <10 ms: 0.11% ± 0.1%) from 7 mice performing the single-whisker detection task (see below) were extracted using MClust (*Redish, 2014*), among which six recordings were from our previous dataset (*Yang et al., 2021*). For *Figure 3*, 5 units from 5 mice were tracked over time (between 1 and 5 days). For *Figure 4*, 68 pupil sessions (34 baseline pupil-matched session pairs) to LC stimulation were acquired from 8 mice, 4 of which were implanted with an optical fiber only (0.39 NA, 200 µm core), and the time between sessions 1 and 2 was 4.4 ± 0.9 days. At the conclusion of the experiments, brains were perfused with PBS followed by 4% paraformaldehyde, postfixed overnight, then cut into 100-µm coronal sections and stained with anti-tyrosine hydroxylase antibody (Thermo Fisher OPA1-04050).

Behavior task was controlled by BControl (C. Brody, Princeton University) or custom-based Arduino hardware and software as described previously (*Yang et al., 2016*; *Yang et al., 2021*; *McBurney-Lin et al., 2020*). In brief, mice were trained to perform a head-fixed, Go/NoGo single-whisker detection task, in which mice reported whether they perceived a brief deflection (0.5 s, 40 Hz or 0.2 s, 25 Hz sinusoidal deflection) to the right C2 whisker by licking toward a water port. A 0.1-s auditory cue (8 kHz tone, ~80 dB SPL) was introduced starting 1–1.5 s before stimulus onset. During all sessions, ambient white noise (cutoff at 40 kHz, ~80 dB SPL) was played through a separate speaker to mask any other potential auditory cues associated with movement of the piezo stimulator. Video of the left pupil (ipsilateral to LC recording and stimulation) was acquired at 50 Hz using a PhotonFocus camera and StreamPix 5 software, or at 20 Hz using a Basler acA1300-200 µm camera and Pylon software. 450 nM blue diode lasers (UltraLasers, MDL-III-450–200 mW) controlled by WaveSurfer (https://www.janelia.org/open-science/wavesurfer) were used for optogenetic stimulation. Electrophysiology, pupil tracking, and optogenetic stimulation were synchronized via a common TTL pulse train. The mating sleeve connecting two ferrules was covered with black tape to prevent light leak. An ambient blue LED was used to constrict the pupil and to mask any potential light leak. <15 mW (RMS) of blue light was measured at the tip of optical fiber. We estimated an excitable volume on the order of 0.05–0.1 mm$^3$ for a 30° cylindrical cone based on 10-mW light power, 2.5 mW/mm$^2$ excitation threshold and 1.4 refractive index of brain tissue (*Boyden et al., 2005*; *Sun et al., 2012*) (brain tissue light transmission calculator: https://web.stanford.edu/group/dlab/cgi-bin/graph/chart.php), containing hundreds of neurons in the LC. Stimulation patterns were delivered every 10–30 s and randomized.

For *Figure 1*, in each recording if the interval between two adjacent spikes was shorter than median inter-spike interval of that unit, the spikes were grouped into a single cluster. Using other time windows (0.1–0.5 s) to group spikes did not affect this analysis for the majority of recordings (data not shown). Peak pupil dilation was defined as the absolute maximum value in a 6-s window following the onset of each cluster (time of the first spike). ROC analysis in *Figure 1f–h* was performed between peak pupil diameter associated with clusters of a given size and number-matched, randomly selected pupil diameter. For *Figure 2*, pupil traces were first smoothed with a 500-ms window to avoid false-positive slope detections. Pupil slopes were then estimated every 200 ms, and a pupil dilation event was defined as the maximum pupil size between sequential positive zero-crossings of the slopes (*Joshi et al., 2016*). For each dilation event, LC spikes were quantified in a −2 to −4 s window from the event. Using a −1 to −3 s window did not affect this analysis (data not shown). Pupil dilation events falling in a bin of 0.3 SD were considered of similar sizes. ROC analysis in *Figure 2e* was performed between LC spike counts associated with pupil dilation events of a similar size and LC spike counts associated with number-matched, randomly selected pupil sizes. ROC analysis in *Figure 2f* was performed the same way as in *Figure 1*. For *Figure 4*, pupil responses in each session were first bootstrapped 100 times with replacement to estimate the mean and confidence interval. Pupil responses to the same optical stimulation were pooled from the two different sessions, and then randomly assigned to session 1 or 2 with replacement. The reported p value represented the proportion of iterations where mean peak pupil responses from the two permutated sessions exceeded the observed difference from 1000 iterations. For *Figure 5*, 9 recordings (out of 13 shown

in *Figure 1e*) from 4 mice during behavior were included with >100 trials and $R^2$ > 0.6. For *Figure 5b*, LC clusters occurring within ±0.5 s from each licking event were excluded from analyzing the pupil– LC relationship as in *Figure 1e*. This window was chosen based on previous results that LC spiking peaked within a few hundred milliseconds of licking onset (*Yang et al., 2021*). For *Figure 4—figure supplement 3*, across-session variability (standard deviation of peak pupil responses) was estimated by resampling trials pooled from all sessions in each condition. The iteration of resampling matched the total number of sessions in that condition. To test whether within-session variability was similar to across-session variability for individual session pairs which exhibited significantly different pupil responses, we first estimated the distribution of across-session variability by resampling trials pooled from both sessions for 1000 iterations and examined whether the variability of individual sessions fell outside 5% of the distribution.

Data were reported as mean ± standard error of the mean unless otherwise noted. We did not use statistical methods to predetermine sample sizes. Sample sizes are similar to those reported in the field. We assigned mice to experimental groups arbitrarily, without randomization or blinding.

## Acknowledgements

We thank Shan Yu, Edward Zagha, Daniel O'Connor, Aaron Seitz, and members of the Yang lab for comments on the manuscript; Laurie Graham for instrument fabrication. This work was supported by UCR startup, UC Regents Faculty Fellowship, Klingenstein-Simons Fellowship Awards in Neuroscience, and NIH grants (R01NS107355, R01NS112200) to HY.

## Additional information

### Funding

| Funder | Grant reference number | Author |
|---|---|---|
| National Institute of Neurological Disorders and Stroke | R01NS107355 | Hongdian Yang |
| National Institute of Neurological Disorders and Stroke | R01NS112200 | Hongdian Yang |
| University of California, Riverside | Startup | Hongdian Yang |
| University of California | Regents Faculty Fellowship | Hongdian Yang |
| Esther A and Joseph Klingenstein Fund | Klingenstein-Simons Fellowship Awards in Neuroscience | Hongdian Yang |

The funders had no role in study design, data collection, and interpretation, or the decision to submit the work for publication.

### Author contributions

Marine Megemont, Jim McBurney-Lin, Conceptualization, Data curation, Formal analysis, Investigation, Methodology, Software, Validation, Visualization, Writing - original draft, Writing - review and editing; Hongdian Yang, Conceptualization, Data curation, Formal analysis, Funding acquisition, Investigation, Methodology, Project administration, Resources, Software, Supervision, Validation, Visualization, Writing - original draft, Writing - review and editing

### Author ORCIDs

Hongdian Yang http://orcid.org/0000-0002-5203-9519

### Ethics

All procedures were performed in accordance with protocols approved by UC Riverside Animal Care and Use Committee (protocol 20190031).

Decision letter and Author response
Decision letter https://doi.org/10.7554/eLife.70510.sa1
Author response https://doi.org/10.7554/eLife.70510.sa2

## Additional files

### Supplementary files
• Transparent reporting form

### Data availability
Source data for main figures (Figures 1-5, MATLAB R2016b files) are uploaded as 'Source data' files.

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
