## [Editor Report]

This work is of interest to those studying neuromodulatory systems, such as the noradrenergic Locus Coeruleus (LC), because it extends prior studies on the relationship between neuronal activity and pupil diameter in awake and anaesthetized animals. Consistent with previous work, the authors show that there is a monotonic relationship between pupil diameter and number of LC spikes, however, they find that pupil diameter can only be used to predict a subset of LC spiking activity levels. This work supports the view that the relationship between pupil diameter and LC spiking may vary dynamically depending on behavioral state.

---

## [Decision Letter]

**Decision letter after peer review:**

[Editors’ note: the authors submitted for reconsideration following the decision after peer review. What follows is the decision letter after the first round of review.]

Thank you for submitting the paper "A graded yet variable relationship between pupil diameter and locus coeruleus activity" for consideration by *eLife*. Your article has been reviewed by 3 peer reviewers, and the evaluation has been overseen by a Reviewing Editor and a Senior Editor. The reviewers have opted to remain anonymous.

We are sorry to say that, after extensive consultation with the reviewers, we have decided that this work will not be considered further for publication by *eLife*.

Although all reviewers agreed that your in vivo spiking data from the LC is impressive and quite valuable, and the attempt to clarify the relationship with pupil measurements is impactful, there were major concerns about several aspects of the paper that preclude publication in *eLife*. Reviewers thought that several conclusions were not strongly supported by the data, including the suggestion that there were two distinct classes of LC neurons and more importantly that variability across cells reflected true variability in the LC-pupil relationship rather than uncontrolled experimental factors. For the optogenetic experiments, the reviewers were not convinced by the limited evidence that the same units were recorded chronically across days, opening up the possibility that variability in the population of neurons recruited by the optogenetic stimulation might be the source of variability in pupil changes. In addition, several key features of the study were not well described, especially the spike clustering method, the anesthesia, and the parameters of the behavioral task.

*Reviewer #1:*

Megemont, McBurney-Lin, and Yang simultaneously recorded pupil diameter and spiking from brainstem noradrenergic locus coeruleus (LC) neurons in awake and anesthetized mice in order to further probe the relationship between LC activity and pupil diameter. Numerous variables are known to correlate with pupil diameter, including behavioral variables (e.g., locomotion, behavioral task-related cognitive processes) and neuronal activity patterns (e.g., cholingergic neurons, inferior colliculus, anterior cingulate cortex). Here, the authors aimed to determine how accurately LC activity can be predicted from pupil diameter, which is a critical question given the relationship between pupil diameter and numerous other variables. To those ends, the authors convincingly show that there is a strong correlation (R^2>0.6) between number of spikes and pupil diameter in 68% of the units (13 out of 19). The authors propose, furthermore, that the activity of only a small proportion of LC units are related to pupil diameter; that specific sub-types of LC neurons have diverse coupling with pupil diameter; that pupil diameter cannot be used to predict the majority of LC spiking activity; and that pupil diameter is not related to LC population firing patterns. However, these latter claims are not supported by the data.

Below is a discussion of the data and authors' claims based on these data, methodological stengths and weaknesses that affect the authors' interpreation of the data, and how the findings relate to prior work:

(1) The authors claim that there is a monotonic relationship between pupil diameter and LC single unit activity. The authors show that is a strong correlation (R^2>0.6) between number of spikes and pupil diameter in 68% of the units (13 out of 19 single units). The data in Figure 1A-E support this claim. However, it is necessary to further show the inter-spike interval distributions and firing rates to demonstrate that the recorded spiking was isolated from single units. These data are qualitatively similar to the prior work cited by the authors reporting trial-by-trial correlations during a behavioral task (Joshi, et al. 2016 and Varazzani, et al. 2015). The authors extend these prior findings of LC-pupil coupling by characterizing the relationship between pupil and LC activity also in the absence of an ongoing behavioral task. Interestingly, while prior work during the behavioral task (Joshi et al., 2016) reported observing a much smaller proportion of LC single units that were significantly correlated with pupil diameter (35% in monkey Oz and 20% in monkey Ci), the task-free LC activity presented here shows the majority of LC units (68%) were coupled with pupil diameter. This is an important finding because it characterizes how the relationship between LC activity and pupil diameter changes whether assessed using the trial structure of a behavioral task versus spontaneous, task-free conditions. Given that the authors found 68% of units coupled to pupil diameter, the authors' proposal that the activity of only a small proportion of LC units are related to pupil diameter is not justified given that the majority were indeed coupled with pupil diameter.

(2) The authors claim that specific sub-types of LC neurons have diverse coupling with pupil diameter. This was substantiated by data showing that there were two subgroups of LC units with different LC firing rates and different coupling to pupil diameter (Figure S2). Here, it would be useful to know how the division was made between subgroups of units. Also, raster examples would help show how a unit can emit a brief cluster of 2 spikes or a brief cluster of 8 spikes, but in both cases spike at ~5Hz. The coupling between pupil dilation and LC unit subgroup does not appear different in Figure S2C, which is borne through by the two-tailed unpaired t-test P=0.09. Finally, the authors interpret their claim that there are 2 subgroups of LC neurons in the context of recent work showing that LC neurons projecting to primary motor cortex have a different firing rate than those projecting to the prefrontal cortex (Chander, et al. 2014) and other work showing that two subsets of LC neurons with different action potential waveform widths fire at different rates (Totah, et al. 2018). However, the comparison to this work and the authors' interpreation is not warrented. First, the difference in firing rates observed between the two subgroups of neurons reported by Chander et al. and by Totah et al. were much smaller than the differences shown in Figure S2. Moreover, the authors did not report the action potential waveform duration in their data to draw a direct comparison with the results of Totah et al.

(3) The authors next test either pupil diameter can be used to infer LC activity to an extent that would allow a direct read-out without actually recording LC neurons. It is well-known, given the number of behavioral variables and activity of various neurons, that such a 1:1 read-out is not possible unless all of these variables were somehow accounted for in the calculation. It is a laudable goal to attempt this work, but the authors do not record from other regions simultaneously with LC neuronal recordings and/or monitor other behavioral variables; therefore, their work cannot be used to assess the potential of pupil diameter as a direct read-out of LC spiking. The data presented by the authors does support clearly a relationship between spontaneous, task-free LC spiking and pupil diameter, but their statement that pupil cannot accurately represent the majority of LC spiking and can only serve as a good proxy for infrequent epochs of high spike rate must be considered in light of the baseline level of LC activity in the recorded mice. Presumably, these data were recorded in a state of "quiet wakefulness." However, if the mice were to enter a drowsy or near-sleep state (which could be quantified with EEG or LFP recordings during these experiments), then LC firing rate would be reduced below what was reported here. Pupil diameter would also be reduced. This greater dynamic range of firing rates and pupil diameter could lead to a stronger relationship between LC spike rate and pupil diameter, even at the lower end of the range of firing rates. Effectively, if the LC spike rate is low and not changing much (quiet wakefulness), then the pupil diameter will be stable at some value and only transient changes of LC spike rate and pupil diameter from that baseline would show correlations; however, if LC rate and pupil diameter are further reduced, then smaller transient changes in rate and diameter would become observable and potentially correlated.

(4) The authors assessed how pupil diameter relates to LC population activity using optogenetic stimulation of the LC. They calculated the volume stimulated and estimate that a few hundred of the ~1,500 mouse LC neurons were synchronously activated. The data show substantial session-to-session variability in the pupil dilation response to optogenetic stimulation which was dissociated from stable responses of the same LC single units recorded across two recording sessions. These data were interpreted as LC activity being largely uncoupled with pupil diameter since the response to optogenetic LC activation fluctuates widely across different recording sessions for only one of those variables (pupil diameter), but the response of the same LC single unit tracked across days does not fluctuate substantially. First, it is particularly important that the authors note that stimulating the LC is not a physiological condition and does not reflect the population activity of the LC, which consists of de-correlated LC neuronal pairs (Totah et al. 2018) and small discrete ensembles (Totah et al. 2020, bioRxiv, https://doi.org/10.1101/2020.03.30.015354). Second, a methodological question that needs clarification in order to assess these data is a discussion and supporting data for how LC neurons were tracked across days. For instance, it has been shown that using wire tetrodes in rodents (as was done in the present study) allows single units to be recorded for typically only 3 days (see Table 1 in Eschenko, O. and Sara, S. J. "Learning-Dependent, Transient Increase of Activity in Noradrenergic Neurons of Locus Coeruleus during Slow Wave Sleep in the Rat: Brain Stem-Cortex Interplay for Memory Consolidation?" Cereb Cortex. (2008) 18, 2596-2603) and, moreover, they were not the same units across days. Thirdly, beyond methodological questions about tracking LC units across days, it is interesting to note that other work that assessed pupil diameter changes in response to LC optogenetic stimulation in both awake and anesthetized mice has shown that there is little variability in pupil diameter response to LC optogenetic stimulation across recording sessions at 2, 4 and 6 week time points (Privitera, M. et al. A complete pupillometry toolbox for real-time monitoring of locus coeruleus activity in rodents. Nat Protoc (2020) 15, 2301-2320). Accordingly, given the methodological questions and also existing work, it is not appropriate to make the strong claim that activation of a large portion of the LC neuronal population is largely uncoupled to pupil diameter.

Overall, given that prior work has shown that the relationship between pupil diameter and LC activity is not 1:1 and that many factors are correlated with pupil diameter, these data provide additional context showing how spontaneous LC activity (no behavioral task) is related to pupil diameter. The impact of such work is strong given that pupil size is often used a non-invasive proxy of neuromodulatory neuronal activity.

*Reviewer #2:*

In this manuscript the authors characterized a graded relationship between locus coeruleus (LC) activity and pupil diameter. This is an extension of their previous work (Yang et al., *eLife*, 2020). Through simultaneously recording of LC activity and pupil diameter, the authors found that pupil diameter increased monotonically with the number of LC spikes, but with substantial variability. Their data also showed that pupil exhibited large session-to-session fluctuations in response to identical optogenetic stimulation of the LC. The variations in the pupil-LC relationship were correlated with all decision related behavioral parameters in mice performing a tactile detection task. The results have very important implications for the use of the pupil diameter as a non-invasive proxy of LC activity, but are perhaps not terribly surprising given previous studies showing pupil diameter is correlated with cortical cholinergic activities.

In this manuscript the authors used simultaneous recordings of LC activity/pupil, optogenetic stimulation and behavior to carefully characterize a graded relationship between LC activity and pupil diameter. The results have very important implications for the use of the pupil size as a non-invasive proxy of LC activity. With that being said, methods were not clearly described in the manuscript and some results may be misinterpreted. Additional LC units should be recorded to ensure the results presented in figure 4 are generalizable. I believe a major revision is necessary before the manuscript being considered publication in the journal.

How to group adjacent spikes into clusters was not clearly defined. It was vaguely described in line 594 but I was confused by the definition, particularly why the clustering depends on each unit's inter-spike-interval. The authors mentioned other time windows (0.1-0.5 s), but it was unclear to me which time window they used in the original analysis.

Figure 1e-h: the authors used linear regression R^2>0.6 as an inclusion criterion for analysis in figure 1e-h. Would the p value of linear regressions a better inclusion criterion since it indicates whether the fit was statistically significant? In addition, it is unclear to me why 0.6 was chosen.

Figure S2c, the p is 0.09 for slope distributions between two subgroups of LC neurons. This indicated that the two subgroups of LC units were not different in coupling with pupil diameter. Therefore, I don't quite understand why the authors argued the two subgroups of LC units coupled with pupil differently (line 99).

Line 139-142: the authors argued here that the same LC unit was recorded across sessions. It is very challenging to record the same neuron across session in chronic settings. The authors should provide more proof to demonstrate that it was the same LC neuron (e.g. waveforms, statistics of spontaneous firing, tail pinch response, etc.).

Line 151: I liked the authors' idea to use anesthetized setup to control for spontaneous pupil fluctuations. However, the authors failed to provide quantitative comparisons of spontaneous pupil fluctuations across awake and anesthetized conditions. In addition, I couldn't find which anesthetics the authors used and how to control for that the difference in evoked pupil responses across sessions was not due to difference in anesthesia levels.

Line 163-165: I don't think results from 9 LC units are able to draw solid conclusions that LC-pupil coupling was strongly correlated with Hit rate, False Alarm rate and decision bias but weakly correlated with overall performance and detection sensitivity. Significantly more LC units (probably 25-30) should be recorded to test if the conclusions are generalizable across LC neurons and animals.

*Reviewer #3:*

The activity of the locus coeruleus (LC) noradrenergic system is thought to play an important role in the regulation of brain states and has been linked to changes in pupil size under constant illumination. However, recent studies, including Yang et al., 2021, have shown that changes in pupil size do not always reliably correlate with LC activity. Here, building on their previous work (Yang et al., 2021), the authors propose to better characterize the relationship between LC activity and pupil dilation, and attempt to explain the source of variability observed in the correlations between these two variables.

Pupillometry, which is an easy-to-use and non-invasive technique, has been used in humans and other animals as a readout of brain states, especially those linked to LC activity. Yet, the neural mechanisms mediating changes in pupil size are complex, and a thorough characterization of the relationship between LC activity and changes in pupil size is still missing in the field. Thus, this paper addresses an important question and should provide relevant information to better understand in which context pupil size can be used to index LC activity.

The data in the paper (pupillometry during a behavioral task paired with extracellular recording of LC neurons and optogenetic stimulation of LC population) are novel and rich and should be adequate to investigate the relationship between pupil size and LC activity. However, some of the analyses performed in this paper are not well designed or properly controlled, and therefore, do not always conclusively support the claims stated in the manuscript.

Some aspects of the paper can be clarified and/or improved:

1) One of the main claims in the paper is that the coupling between LC activity and pupil dilation is variable and that this variability is dependent on specific states (i.e., the decision-bias of mice performing a go/no-go task). Yet, pupil size in awake animals, even under constant luminance and in the absence of locomotion, remains intrinsically variable. In fact, facial movements such as whisking or licking (that are also involved in this go/no-go task) have been associated with pupil dilation. Therefore, different types of movements, as well as their rate and vigor, could potentially account for some of the variability observed in the correlation between pupil and NC activity. Yet, there is no mention of such confounds in the paper.

2) Most of the analyses and conclusions of the paper are based on the graded relationship between LC activity and change in pupil size quantified in Figure 1. This relationship relies on the size of spike clusters identified in the activity of single LC neurons and on the maximum change in pupil size within a 6s window after the first spike of a given cluster. However, it is not stated in the paper whether it was ensured that no other clusters occurred within the 6s window. If this condition is not respected, the number of spikes occurring in the time window of interest may themselves be variable, and therefore, not necessarily consistent with the cluster sizes used in the analysis.

3) It is briefly mentioned in Figure S2 that subgroups of neurons have instantaneous firing rate changing differently with cluster size. Interestingly, these subgroups have different coupling with pupil dilation. Thus, it is possible that changes in instantaneous firing rate, rather than cluster size, more reliably correlate with pupil dilation, and in turn, explain away some of the variability that is observed with cluster size. This alternative has not been explored in the paper.

4) In Figure 3, the reliability of the effect of optogenetic stimulation on the population of LC activity is inferred from one single unit across two sessions. Even if one single neuron appears stable between two sessions, it is not unlikely that the population activity varies from session to session. Thus, the data reported in this figure do not convincingly support a consistent transduction of optical stimulation to LC population activity.

5) Similarly, variability of pupil responses from "session-to-session", is quantified across only two sessions (i.e., two points), in just a couple of animals. From one session to the next, some experimental parameters that cannot be controlled for may change and add variability to the stimulations and measurements. Even in anesthetized animals, the level of anesthesia could differ from one session to the next. Thus, to more convincingly describe variability across sessions, one would need to add more sessions. Alternatively, the results could be compared to a "baseline variability" quantified from matched non-stimulated trials or from stimulated trials in control mice.

6) It is stated that pupil dilation is a good proxy for LC activity only for infrequent (<10%) and strong activity events. However, the authors do not comment on the nature of these infrequent events, especially in relation to the decision-bias.

Figure 1:

1) During the first read, I had assumed that Figure 1 showed recordings from sessions where mice were awake but passive. After reaching Figure 4, I then became unclear about whether mice were performing the go/no-go task already in Figure 1.

2) It is not specified whether the peak pupil size occurs at a consistent latency after the first cluster spike or whether the latency is cluster size dependent.

3) A different way to assess the correlation between pupil and LC activity, as well as the latency of the pupil response, is to first compute a smooth instantaneous firing rate signal (convolved with some gaussian kernels for instance), and then perform a cross-correlation between the firing rate and the pupil signal. This analysis will directly reveal the strength of the correlation and lag between the two vectors. Using the instantaneous firing rate could be informative especially with regards to the results in Figure S2.

4) Related to comment 1 of the public review: here the authors may want to condition their analysis to epochs when mice are licking, whisking, etc. (assuming that these data have been recorded from the behavioral task). If there are also some video recordings of the face (not just the pupil), the authors can even track facial movements and correlate such signals with the pupil response and LC activity.

Figure 3:

1) The authors claim that they have recorded the same unit across different days. Yet, it is not described in the text nor in the methods what criteria are used to define those units as identical.

2) I appreciate that examples of single mice are shown (Figure 3, Figure S8-10), but it would also help to see a summary plot with all the mice, all the sessions, and all the different conditions.

3) Related to the point above: assuming that there are significant session-to-session fluctuations in the pupil response to identical LC stimulation, could you describe it further? Does the stimulation of LC always induce a smaller pupil response on the second day? Is it consistent across mice?

4) The 95% confidence intervals seem extremely small compared to their respective P-values.

Figure 4:

1) This manuscript is formatted as a Research Advances and builds on Yang et al., 2021. Yet, it would help to have a sentence or two to briefly describe the task, either here or in the methods.

2) It is my understanding that the data shown here are from the same units as the data Figure 1e. However, it is unclear whether the results come from the same sessions. If they do, why are the slopes in Figure 4 not the same as the slope in Figure 1e (right)?

3) The more impulsive the animal (negative bias), the stronger the coupling between pupil and LC. Therefore, here again, we cannot rule out the implication of movements, since impulsive mice may tend to respond stronger and faster (larger false alarm, larger hit rate, which also correlates with stronger pupil-LC coupling). Without this basic control for movement, it is difficult to believe that the LC-pupil relationship directly depends on cognitive states such as motivation and engagement as stated in the paper, rather than difference in movements.

[Editors’ note: further revisions were suggested prior to acceptance, as described below.]

Thank you for resubmitting your work entitled "A graded yet variable relationship between pupil diameter and locus coeruleus activity" for further consideration by *eLife*. Your revised article has been evaluated by Joshua Gold (Senior Editor) and a Reviewing Editor.

The manuscript has been improved but there are some remaining issues that need to be addressed, as outlined below:

The new analyses and more tempered conclusions make the study more convincing compared to the previous version. However, there are remaining methodological and analysis concerns that should be addressed. If these concerns are addressed, there is agreement that the revised study would contribute to the field of LC physiology and how LC activity relates to pupil size, and would be suitable for publication in *eLife*.

Please address all the concerns outlined by the reviewers below.

In addition, it seems like it would be possible to perform an analysis of variability of optostim-to-dilation *within* each session and show that this variability is similar to the variability across sessions. This would help ameliorate concerns about variability in the LC-pupil relationship being due to variability in the long term stability of recordings and the optical fiber across sessions. The authors should provide information about the within-session vs cross-session variability.

*Reviewer #1:*

The authors main claim is that pupil diameter is not an accurate real-time readout of locus coeruleus activity. With the revised text, figures, and new analyses, the claim is convincing. I think the title change more accurately conveys the findings of the study.

With the addition of new analyses, the authors convincingly show that there is substantial session-to-session variability in the pupil dilation response to optogenetic stimulation (Figure 4C, D, etc.). In response to the prior review, the authors addressed the differences in relation to Privitera, et al. (2020) regarding pupil response variability across sessions. Moreover, with the addition of a new analysis of waveform correlations for LC single units across days (Figure 3G) the authors convincingly show that the same units can be tracked across days. The authors then point out that, in contrast to the pupil response to opto-stim having variable responses across days, the response of the same unit to opto-stim on different days is not variable (Figure 3H). Based on these data, the authors suggest that LC neurons are not an "accurate real-time readout of locus coeruleus activity". I think these new analyses and the data collectively do support this statement.

There is one caveat to this, which is that different size populations of LC neurons could be driven by optical stimulation on different days. This could occur because slow changes in the tissue over days (gliosis, blood flow, inflammation) could occlude light or alter light transmission. Under this scenario, the neurons tracked across days were likely not affected by changes in inflammation/gliosis (which is why they could be tracked), but this does not mean that LC neurons which could not be monitored by the electrode did not receive different light transmission on different days. If different size populations were stimulated, then the overall pupil dilation might change. This caveat is of course nothing that the authors could address given the current technology limitations and it is mentioned just for their consideration and could be discussed if they agree and find this warranted.

There is a question about the number of units in Figure 3H. The legend states 5 awake units (3H, left panel) and does not report the number of units for anesthetized (3H, right panel). However, it appears that there are only 4 awake units (dots) in the left panel.

There remains one methodological concern regarding single unit recordings.

The authors added additional quantification of spike train and spike clustering characteristics. The new Figure S1A shows that the inter-spike interval (ISI) histogram has most of its density around a mean of 20-25 msec. However, the figure legend reports a median of 200 msec. That median does not appear to match the distribution plotted, given that nearly the entire distribution plotted in Figure S1A is <0.2 sec. In prior work that recorded awake monkey LC single units, the ISI histogram has been shown to peak between ~150 msec and 350 msec with ISI's <100 msec rarely being observed (Aston-Jones, G., Rajkowski, J., Kubiak, P. and Alexinsky, T. Locus coeruleus neurons in monkey are selectively activated by attended cues in a vigilance task. J Neurosci 14, 4467-4480 (1994)). Other studies of multi-unit activity have found a median ISI of 53 msec with an inter-quartile range of 20 to 157 msec (Kalwani, R. M., Joshi, S. and Gold, J. I. Phasic activation of individual neurons in the locus ceruleus/subceruleus complex of monkeys reflects rewarded decisions to go but not stop. J Neurosci 34, 13656 13669 (2014)). This range appears to be in line with the authors' presentation of the ISI distribution in Figure S1A. On the other hand, the example waveforms look convincing. Further quantification is necessary, such as showing some ISI histograms for individual single units as well as some waveform clustering examples and average waveforms. Given the small number of units in the study, I think it would be helpful for the field to see for each unit the average waveform and the ISI histogram for each unit in a supplementary figure. This can help the field to better interpret their work in the context of others. Additionally, a presentation of the distribution of waveform durations against firing rate would also provide helpful support for interpreting the results.

While the authors' claim that pupil diameter is not an accurate real-time readout of locus coeruleus activity is supported by their data, I am still not convinced that this is not already a known and accepted idea. The field clearly knows that many different neuronal circuits affect pupil size. For example, it is known that LC output (axonal activity) tracks specific aspects of pupil dynamics but cholinergic neurons another aspect (Reimer, J. et al. Pupil fluctuations track rapid changes in adrenergic and cholinergic activity in cortex. Nat Commun 7, 13289 (2016)). With that being said, I think these data support their claim and add to the literature on this topic.

*Reviewer #3:*

The authors have performed new analyses (Figure 3, 4, S1, S2c, S8, S9) in an attempt to address most of the reviewers' comments. Missing information in the text and methods has been added, especially regarding the optogenetics experiments, and the discussion has been extended. This helps clarify some aspects of the paper. Some of the results are now more convincing (especially Figure 3 and 4) but other analyses remain oversimplified and therefore do not strongly support several conclusions.

1) In Figure 1 and Figure S2, depending on the method used to quantify the relationship between pupil dilation and cluster size, a different number of neurons meet the R2>0.6 criterion. This seems problematic since following analyses only focus on this subset of neurons. Are there always the same neurons that consistently fall below the 0.6 criterion? Do the results hold when all the recorded neurons are included?

2) I am still not fully convinced about the validity of the pupil dilation vs. cluster size analysis. Yet, this result (and its variability) is at the essence of the main conclusion of the paper. In particular, this analysis assumes that discrete events (spike clusters) linearly increase pupil size after a fixed latency (2-3s from Yang et al., 2021). Not accounting for the number of spikes occurring after a given cluster in the 3 s or 6 s time window disregard the fact that the latency may be variable or that the relationship between LC spiking and changes in pupil size could be nonlinear. This could reflect true variability in the LC-pupil relationship.

3) In Figure 5b, the non-liking period is defined as the activity outside a one second window around each licking events (clusters occurring within {plus minus}0.5 s from a lick were excluded). This choice is somewhat arbitrary, or at least not well motivated in the text. Besides, the pupil response is slow and the authors mentioned that the latency between peak pupil dilation and LC spikes is around 2 to 3 s. Therefore, the 1 s window may not be the most appropriate. A more convincing approach would be to condition the analysis on the number of licks performed in a trial for instance.

---

## [Author Response]

[Editors’ note: The authors appealed the original decision. What follows is the authors’ response to the first round of review.]

Reviewer #1:Megemont, McBurney-Lin, and Yang simultaneously recorded pupil diameter and spiking from brainstem noradrenergic locus coeruleus (LC) neurons in awake and anesthetized mice in order to further probe the relationship between LC activity and pupil diameter. Numerous variables are known to correlate with pupil diameter, including behavioral variables (e.g., locomotion, behavioral task-related cognitive processes) and neuronal activity patterns (e.g., cholingergic neurons, inferior colliculus, anterior cingulate cortex). Here, the authors aimed to determine how accurately LC activity can be predicted from pupil diameter, which is a critical question given the relationship between pupil diameter and numerous other variables. To those ends, the authors convincingly show that there is a strong correlation (R^2>0.6) between number of spikes and pupil diameter in 68% of the units (13 out of 19). The authors propose, furthermore, that the activity of only a small proportion of LC units are related to pupil diameter; that specific sub-types of LC neurons have diverse coupling with pupil diameter; that pupil diameter cannot be used to predict the majority of LC spiking activity; and that pupil diameter is not related to LC population firing patterns. However, these latter claims are not supported by the data.Below is a discussion of the data and authors' claims based on these data, methodological stengths and weaknesses that affect the authors' interpreation of the data, and how the findings relate to prior work:(1) The authors claim that there is a monotonic relationship between pupil diameter and LC single unit activity. The authors show that is a strong correlation (R^2>0.6) between number of spikes and pupil diameter in 68% of the units (13 out of 19 single units). The data in Figure 1A-E support this claim. However, it is necessary to further show the inter-spike interval distributions and firing rates to demonstrate that the recorded spiking was isolated from single units. These data are qualitatively similar to the prior work cited by the authors reporting trial-by-trial correlations during a behavioral task (Joshi, et al. 2016 and Varazzani, et al. 2015). The authors extend these prior findings of LC-pupil coupling by characterizing the relationship between pupil and LC activity also in the absence of an ongoing behavioral task. Interestingly, while prior work during the behavioral task (Joshi et al., 2016) reported observing a much smaller proportion of LC single units that were significantly correlated with pupil diameter (35% in monkey Oz and 20% in monkey Ci), the task-free LC activity presented here shows the majority of LC units (68%) were coupled with pupil diameter. This is an important finding because it characterizes how the relationship between LC activity and pupil diameter changes whether assessed using the trial structure of a behavioral task versus spontaneous, task-free conditions. Given that the authors found 68% of units coupled to pupil diameter, the authors' proposal that the activity of only a small proportion of LC units are related to pupil diameter is not justified given that the majority were indeed coupled with pupil diameter.

We included new Figure S1a and b to show individual and group inter-spike interval distributions, respectively. Additional quality control measures are now included in Methods, line 667-669:

“Nineteen single unit recordings (cluster quality measure L_ratio_: 0.01 ± 0.005; firing rate: 1.65 ± 0.25 spikes/s; percent ISI < 10 ms: 0.11% ± 0.1%) from seven mice performing the singlewhisker detection task (see below) were extracted using MClust (Redish, 2014)…”

68% of units showed monotonic relationship with pupil diameter. However, this relationship does not guarantee that pupil diameter can be used to predict LC activity on a moment-by-moment basis. We now make this point more clearly throughout the manuscript, e.g., line 99-100:

“Although pupil diameter exhibited an overall monotonic relationship with LC spiking, it does not necessarily warrant pupil diameter being an accurate readout of LC activity.”

We also changed the manuscript title to ‘Pupil diameter is not an accurate real-time readout of locus coeruleus activity’ to reflect this point.

We apologize for the misunderstanding. All LC-pupil recordings shown in Figure 1, 2 and 5 (old Figure 4) were acquired during behavior. We now made that clear in the manuscript, e.g., line 68-79:

“To address these questions, we recorded spiking activity from optogenetically-tagged LC neurons simultaneously with pupil diameter in head-fixed mice trained to perform a tactile detection task…”

(2) The authors claim that specific sub-types of LC neurons have diverse coupling with pupil diameter. This was substantiated by data showing that there were two subgroups of LC units with different LC firing rates and different coupling to pupil diameter (Figure S2). Here, it would be useful to know how the division was made between subgroups of units. Also, raster examples would help show how a unit can emit a brief cluster of 2 spikes or a brief cluster of 8 spikes, but in both cases spike at ~5Hz. The coupling between pupil dilation and LC unit subgroup does not appear different in Figure S2C, which is borne through by the two-tailed unpaired t-test P=0.09. Finally, the authors interpret their claim that there are 2 subgroups of LC neurons in the context of recent work showing that LC neurons projecting to primary motor cortex have a different firing rate than those projecting to the prefrontal cortex (Chander, et al. 2014) and other work showing that two subsets of LC neurons with different action potential waveform widths fire at different rates (Totah, et al. 2018). However, the comparison to this work and the authors' interpreation is not warrented. First, the difference in firing rates observed between the two subgroups of neurons reported by Chander et al. and by Totah et al. were much smaller than the differences shown in Figure S2. Moreover, the authors did not report the action potential waveform duration in their data to draw a direct comparison with the results of Totah et al.

All reviewers raised a similar point regarding the results in the old Figure S2. Although in the original manuscript we stated that the two groups of LC units ‘appeared to be coupled with pupil differently’, we completely agree that the results are not conclusive due to weak statistics. Therefore, this part of the results has been removed. In addition, we performed the Bayesian analysis suggested by this reviewer (recommendation #5) and the Bayes Factor (BF10) was 1.24, indicating inconclusive evidence.

(3) The authors next test either pupil diameter can be used to infer LC activity to an extent that would allow a direct read-out without actually recording LC neurons. It is well-known, given the number of behavioral variables and activity of various neurons, that such a 1:1 read-out is not possible unless all of these variables were somehow accounted for in the calculation. It is a laudable goal to attempt this work, but the authors do not record from other regions simultaneously with LC neuronal recordings and/or monitor other behavioral variables; therefore, their work cannot be used to assess the potential of pupil diameter as a direct read-out of LC spiking. The data presented by the authors does support clearly a relationship between spontaneous, task-free LC spiking and pupil diameter, but their statement that pupil cannot accurately represent the majority of LC spiking and can only serve as a good proxy for infrequent epochs of high spike rate must be considered in light of the baseline level of LC activity in the recorded mice. Presumably, these data were recorded in a state of "quiet wakefulness." However, if the mice were to enter a drowsy or near-sleep state (which could be quantified with EEG or LFP recordings during these experiments), then LC firing rate would be reduced below what was reported here. Pupil diameter would also be reduced. This greater dynamic range of firing rates and pupil diameter could lead to a stronger relationship between LC spike rate and pupil diameter, even at the lower end of the range of firing rates. Effectively, if the LC spike rate is low and not changing much (quiet wakefulness), then the pupil diameter will be stable at some value and only transient changes of LC spike rate and pupil diameter from that baseline would show correlations; however, if LC rate and pupil diameter are further reduced, then smaller transient changes in rate and diameter would become observable and potentially correlated.

We completely agree with the comment that ‘given the number of behavioral variables and activity of various neurons, that such a 1:1 read-out is not possible unless all of these variables were somehow accounted for in the calculation’. However, it is our impression that this is not everyone’s understanding. Actually, the main message we aimed to convey in this manuscript is, in a quantitative manner, that pupil is not a 1:1 readout of LC in most circumstances. We now include more extensive discussion on this topic, e.g., line 238-247:

“How may brain states modulate pupil-LC coupling? Pupil size changes have been linked to activity in other brain areas and neuromodulatory systems, including the medial prefrontal cortex, the inferior colliculus and cholinergic signaling (Joshi et al., 2016; Reimer et al., 2016; Okun et al., 2019; Kucyi and Parvizi, 2020; Pais-Roldán et al., 2020; Sobczak et al., 2021). A recent study found that pupil responses to dorsal raphe stimulation exhibited task uncertainty dependent variations (Cazettes et al., 2020). Therefore, it is possible that in high motivation/engagement states, multiple circuits including the LC synergistically influence pupil size changes, resulting in the apparently stronger pupil-LC coupling. Future experiments are needed to elucidate how pupil and LC interact with these brain circuits during different behavioral contexts and cognitive processes.”

Again, we apologize for the misunderstanding (response to #1). All LC-pupil recordings (Figure 1, 2, 5) were acquired during behavior. Our new Figure 5b supports that pupil-LC coupling is correlated with the state of the mice, which cannot be accounted for by the differences in licking. We have included new discussion on this topic, line 232-236:

“Although mice licked more during high motivation or high engagement trials (Berditchevskaia et al., 2016; Allen et al., 2019; McBurney-Lin et al., 2020), our data show that licking alone cannot account for the tight correlation between the variations of the behavioral variables and the variations in the pupil-LC relationship (Figure 5), suggesting that pupil-LC coupling is brain state dependent.”

Line 249-254:

“Another possibility is that higher engagement states may be intimately associated with more ‘uninstructed’ movements as revealed by recent work (Musall et al., 2019), which can drive robust neuronal activity throughout the brain (Musall et al., 2019; Steinmetz et al., 2019; Stringer et al., 2019; Salk_off_ et al., 2020). Future studies with comprehensive movement monitoring will determine whether more frequent movements, both task-related and task-unrelated, during periods of high motivation/engagement underlie the stronger pupil-LC coupling.”

(4) The authors assessed how pupil diameter relates to LC population activity using optogenetic stimulation of the LC. They calculated the volume stimulated and estimate that a few hundred of the ~1,500 mouse LC neurons were synchronously activated. The data show substantial session-to-session variability in the pupil dilation response to optogenetic stimulation which was dissociated from stable responses of the same LC single units recorded across two recording sessions. These data were interpreted as LC activity being largely uncoupled with pupil diameter since the response to optogenetic LC activation fluctuates widely across different recording sessions for only one of those variables (pupil diameter), but the response of the same LC single unit tracked across days does not fluctuate substantially. First, it is particularly important that the authors note that stimulating the LC is not a physiological condition and does not reflect the population activity of the LC, which consists of de-correlated LC neuronal pairs (Totah et al. 2018) and small discrete ensembles (Totah et al. 2020, bioRxiv, https://doi.org/10.1101/2020.03.30.015354). Second, a methodological question that needs clarification in order to assess these data is a discussion and supporting data for how LC neurons were tracked across days. For instance, it has been shown that using wire tetrodes in rodents (as was done in the present study) allows single units to be recorded for typically only 3 days (see Table 1 in Eschenko, O. and Sara, S. J. "Learning-Dependent, Transient Increase of Activity in Noradrenergic Neurons of Locus Coeruleus during Slow Wave Sleep in the Rat: Brain Stem-Cortex Interplay for Memory Consolidation?" Cereb Cortex. (2008) 18, 2596-2603) and, moreover, they were not the same units across days. Thirdly, beyond methodological questions about tracking LC units across days, it is interesting to note that other work that assessed pupil diameter changes in response to LC optogenetic stimulation in both awake and anesthetized mice has shown that there is little variability in pupil diameter response to LC optogenetic stimulation across recording sessions at 2, 4 and 6 week time points (Privitera, M. et al. A complete pupillometry toolbox for real-time monitoring of locus coeruleus activity in rodents. Nat Protoc (2020) 15, 2301-2320). Accordingly, given the methodological questions and also existing work, it is not appropriate to make the strong claim that activation of a large portion of the LC neuronal population is largely uncoupled to pupil diameter.

We thank the reviewer for these comments. We discuss the potential weakness of LC optical stimulation, line 216-219:

“However, optogenetic stimulation tends to synchronize neuronal activity, which may not reflect the physiological condition (Totah et al., 2018). Future experiments with the ability to simultaneously record from multiple opto-tagged LC neurons will further investigate the relationship between pupil diameter and population-level LC activity.”

We have provided new data to support that the putatively same units were tracked, based on opto-tagging, spike clustering and waveform comparison (Figure 3c-g).

We have included new data to show group pupil responses (Figure 4c-h). Pupil responses in an earlier session were not consistently higher or lower than a later session. Therefore, such session-to-session fluctuations were not observable from group data (Figure S8), which is consistent with Privitera et al., 2020. Line 144-149:

“Group data from multiple mice further demonstrated that significant session-to-session fluctuations of pupil responses were prevalent but not directional (solid lines in Figure 4c, d), i.e., pupil responses in an earlier session (session 1) were not consistently higher or lower than in a later session (session 2). Therefore, such session-to-session fluctuations were not observable from group comparisons (Figure S8, Privitera et al., 2020).”

In addition, some of the new pupil data were acquired a few hours apart (magenta arrows in Figure 4c-f) and still had substantial differences. In our opinion, it further alleviates the concern that LC responses may change over time, presumably due to slow drift.

Reviewer #2:In this manuscript the authors characterized a graded relationship between locus coeruleus (LC) activity and pupil diameter. This is an extension of their previous work (Yang et al., eLife, 2020). Through simultaneously recording of LC activity and pupil diameter, the authors found that pupil diameter increased monotonically with the number of LC spikes, but with substantial variability. Their data also showed that pupil exhibited large session-to-session fluctuations in response to identical optogenetic stimulation of the LC. The variations in the pupil-LC relationship were correlated with all decision related behavioral parameters in mice performing a tactile detection task. The results have very important implications for the use of the pupil diameter as a non-invasive proxy of LC activity, but are perhaps not terribly surprising given previous studies showing pupil diameter is correlated with cortical cholinergic activities.In this manuscript the authors used simultaneous recordings of LC activity/pupil, optogenetic stimulation and behavior to carefully characterize a graded relationship between LC activity and pupil diameter. The results have very important implications for the use of the pupil size as a non-invasive proxy of LC activity. With that being said, methods were not clearly described in the manuscript and some results may be misinterpreted. Additional LC units should be recorded to ensure the results presented in figure 4 are generalizable. I believe a major revision is necessary before the manuscript being considered publication in the journal.How to group adjacent spikes into clusters was not clearly defined. It was vaguely described in line 594 but I was confused by the definition, particularly why the clustering depends on each unit's inter-spike-interval. The authors mentioned other time windows (0.1-0.5 s), but it was unclear to me which time window they used in the original analysis.

We now described in more details in Methods, line 700-701:

“For Figure 1, in each recording if the interval between two adjacent spikes was shorter than median inter-spike interval of that unit, the spikes were grouped into a single cluster.

Each unit was clustered by its own inter-spike interval (ITI) because units had different ITIs. We include a new Figure S1a, b to show the distribution of inter-spike interval.

Figure 1e-h: the authors used linear regression R^2>0.6 as an inclusion criterion for analysis in figure 1e-h. Would the p value of linear regressions a better inclusion criterion since it indicates whether the fit was statistically significant? In addition, it is unclear to me why 0.6 was chosen.

We performed this analysis and found that all included recordings (R^2^>0.6) had significant p values. None of the excluded recordings except one had significant p value. We now include a new Figure S1c to show the distribution of R^2^ for all recordings, illustrating that the group with high R^2^ values was well separated, and that setting the threshold between 0.5 and 0.8 as the inclusion criterion won’t affect any following results.

Figure S2c, the p is 0.09 for slope distributions between two subgroups of LC neurons. This indicated that the two subgroups of LC units were not different in coupling with pupil diameter. Therefore, I don't quite understand why the authors argued the two subgroups of LC units coupled with pupil differently (line 99).

We agree with this point. Please see our response to Reviewer1 comment #2 and recommendation #5. We have removed this part of the results due to weak statistics and inconclusive evidence.

Line 139-142: the authors argued here that the same LC unit was recorded across sessions. It is very challenging to record the same neuron across session in chronic settings. The authors should provide more proof to demonstrate that it was the same LC neuron (e.g. waveforms, statistics of spontaneous firing, tail pinch response, etc.).

Please see our response to Reviewer1 #4. We now provide new data to support that the putatively same units were tracked over days, based on opto-tagging, spike clustering and waveform comparison (Figure 3c-g).

Line 151: I liked the authors' idea to use anesthetized setup to control for spontaneous pupil fluctuations. However, the authors failed to provide quantitative comparisons of spontaneous pupil fluctuations across awake and anesthetized conditions. In addition, I couldn't find which anesthetics the authors used and how to control for that the difference in evoked pupil responses across sessions was not due to difference in anesthesia levels.

We thank the reviewer for this comment. We have performed new analysis to show that the occurrence and amplitude of spontaneous pupil dilation events were reduced in the anesthetized condition (Figure S9). We always used 2% isoflurane for anesthetized experiments. This is now included in the manuscript.

Line 163-165: I don't think results from 9 LC units are able to draw solid conclusions that LC-pupil coupling was strongly correlated with Hit rate, False Alarm rate and decision bias but weakly correlated with overall performance and detection sensitivity. Significantly more LC units (probably 25-30) should be recorded to test if the conclusions are generalizable across LC neurons and animals.

We agree with the reviewer that in order to make such comparisons (i.e., stronger vs. weaker correlation), more recordings will be needed. The main message we wanted to convey here is that the variations in pupil-LC coupling is correlated with behavioral variables. As a result, in the revised manuscript, we do not make such comparisons and only draw the conclusion that decision-biased related variables can explain the variations of pupil-LC coupling (Figure 5). In the meantime, we want to emphasize that making recordings from opto-tagged LC neurons during behavior presents a major challenge to the field and has rarely been done.

Reviewer #3:The activity of the locus coeruleus (LC) noradrenergic system is thought to play an important role in the regulation of brain states and has been linked to changes in pupil size under constant illumination. However, recent studies, including Yang et al., 2021, have shown that changes in pupil size do not always reliably correlate with LC activity. Here, building on their previous work (Yang et al., 2021), the authors propose to better characterize the relationship between LC activity and pupil dilation, and attempt to explain the source of variability observed in the correlations between these two variables.Pupillometry, which is an easy-to-use and non-invasive technique, has been used in humans and other animals as a readout of brain states, especially those linked to LC activity. Yet, the neural mechanisms mediating changes in pupil size are complex, and a thorough characterization of the relationship between LC activity and changes in pupil size is still missing in the field. Thus, this paper addresses an important question and should provide relevant information to better understand in which context pupil size can be used to index LC activity.The data in the paper (pupillometry during a behavioral task paired with extracellular recording of LC neurons and optogenetic stimulation of LC population) are novel and rich and should be adequate to investigate the relationship between pupil size and LC activity. However, some of the analyses performed in this paper are not well designed or properly controlled, and therefore, do not always conclusively support the claims stated in the manuscript.Some aspects of the paper can be clarified and/or improved:1) One of the main claims in the paper is that the coupling between LC activity and pupil dilation is variable and that this variability is dependent on specific states (i.e., the decision-bias of mice performing a go/no-go task). Yet, pupil size in awake animals, even under constant luminance and in the absence of locomotion, remains intrinsically variable. In fact, facial movements such as whisking or licking (that are also involved in this go/no-go task) have been associated with pupil dilation. Therefore, different types of movements, as well as their rate and vigor, could potentially account for some of the variability observed in the correlation between pupil and NC activity. Yet, there is no mention of such confounds in the paper.

The reviewer raised an excellent point. In general, we think these are emerging and important questions in the field: what is the relationship between brain states and movements? How could different types of movements affect physiological processes, including pupil-LC coupling? Unfortunately, we do not have whisker tracking in the current dataset. Given that licking is a major type of movement during the task, we performed new analysis to show that licking cannot account for the variability (Figure 5b). However, the reviewer is correct that other types of movements may come into play. We include new sections to discuss this potential confound, line 232-236:

“Although mice licked more during high motivation or high engagement trials (Berditchevskaia et al., 2016; Allen et al., 2019; McBurney-Lin et al., 2020), our data show that licking alone cannot account for the tight correlation between the variations of the behavioral variables and the variations in the pupil-LC relationship (Figure 5), suggesting that pupil-LC coupling is brain state dependent.”

Line 249-254:

“Another possibility is that higher engagement states may be intimately associated with more ‘uninstructed’ movements as revealed by recent work (Musall et al., 2019), which can drive robust neuronal activity throughout the brain (Musall et al., 2019; Steinmetz et al., 2019; Stringer et al., 2019; Salk_off_ et al., 2020). Future studies with comprehensive movement monitoring will determine whether more frequent movements, both task-related and task-unrelated, during periods of high motivation/engagement underlie the stronger pupil-LC coupling.”

2) Most of the analyses and conclusions of the paper are based on the graded relationship between LC activity and change in pupil size quantified in Figure 1. This relationship relies on the size of spike clusters identified in the activity of single LC neurons and on the maximum change in pupil size within a 6s window after the first spike of a given cluster. However, it is not stated in the paper whether it was ensured that no other clusters occurred within the 6s window. If this condition is not respected, the number of spikes occurring in the time window of interest may themselves be variable, and therefore, not necessarily consistent with the cluster sizes used in the analysis.

Our previous work shows that the latency between peak pupil dilation and LC spikes is around 23 s (Yang et al., 2021, *eLife*). We chose the 6-s window to account for the long latency between pupil responses and large LC clusters (For example, Figure 1d, 6-8 spikes panels). We performed new analysis using a shorter window (3-s) and found that it did not affect the pupil-LC relationship (Figure S2c), and the results in Figure 5 were not affected using the 3-s window (not shown). We can include the latter result if deemed critical.

3) It is briefly mentioned in Figure S2 that subgroups of neurons have instantaneous firing rate changing differently with cluster size. Interestingly, these subgroups have different coupling with pupil dilation. Thus, it is possible that changes in instantaneous firing rate, rather than cluster size, more reliably correlate with pupil dilation, and in turn, explain away some of the variability that is observed with cluster size. This alternative has not been explored in the paper.

As described above, results related to old Figure S2 have been removed due to weak statistics and inconclusive evidence. However, we performed the suggested analysis and found that instantaneous firing rate of the clusters did not better correlate with pupil diameter. We can include this result if deemed critical.

4) In Figure 3, the reliability of the effect of optogenetic stimulation on the population of LC activity is inferred from one single unit across two sessions. Even if one single neuron appears stable between two sessions, it is not unlikely that the population activity varies from session to session. Thus, the data reported in this figure do not convincingly support a consistent transduction of optical stimulation to LC population activity.

We have provided new data to support that the putatively same units were tracked (Figure 3c-g), and all units responded similarly across sessions (Figure 3h). In our opinion, being able to track the putatively same units across days strongly suggest that the tetrodes and optical fiber had minimal shift inside the tissue (< tens of micrometers). Given that the estimated excitable volume is on the order of 0.1 mm^3^, it is the largely same population of neurons being stimulated. The combined evidence that (1) the largely same population of neurons were activated and (2) all putatively same neurons responded similarly across sessions strongly suggest that population responses were similar. In contrast, pupil responses can be dramatically different (e.g., doubled or halved, Figure 4). Furthermore, some of the new pupil data were acquired a few hours apart (magenta arrows in Figure 4c-f) and still had substantial differences. In our opinion, it further alleviates the concern that LC population responses could be different from session to session, presumably due to slow drift over time. However, due to technical limitations, we are not able to obtain multiple opto-tagged LC units simultaneously, and to our knowledge, no published work has accomplished this. This is now discussed, line 216-219:

“However, optogenetic stimulation tends to synchronize neuronal activity, which may not reflect the physiological condition (Totah et al., 2018). Future experiments with the ability to simultaneously record from multiple opto-tagged LC neurons will further investigate the relationship between pupil diameter and population-level LC activity.”

5) Similarly, variability of pupil responses from "session-to-session", is quantified across only two sessions (i.e., two points), in just a couple of animals. From one session to the next, some experimental parameters that cannot be controlled for may change and add variability to the stimulations and measurements. Even in anesthetized animals, the level of anesthesia could differ from one session to the next. Thus, to more convincingly describe variability across sessions, one would need to add more sessions. Alternatively, the results could be compared to a "baseline variability" quantified from matched non-stimulated trials or from stimulated trials in control mice.

We thank the reviewer for the comments. For consistency, we always used 2% isoflurane in anesthetized recordings. Session-to-session pupil responses were quantified in two sessions because these sessions were baseline-matched, i.e., they had very similar baseline pupil diameter, as it is known that evoked pupil sizes are influenced by baseline pupil. We have included new group data of baseline-matched pupil responses to optical stimulation from more sessions and more mice to support our conclusions (Figure 4c-h).

6) It is stated that pupil dilation is a good proxy for LC activity only for infrequent (<10%) and strong activity events. However, the authors do not comment on the nature of these infrequent events, especially in relation to the decision-bias.

We have included new discussions on these points. Line 195-200:

“Large pupil or LC responses have been reported to correlate with a variety of task-related processes, including sensory cue, decision formation, positive feedback, choice bias and action (Rajkowski et al., 1994; Usher et al., 1999; Kalwani et al., 2014b; Bouret and Richmond, 2015; de Gee et al., 2017, 2020; Schriver et al., 2020; Yang et al., 2021). In light of this, our work suggests that the infrequent but strong pupil dilation events can be used as an accurate inference of LC activation in response to sensory stimuli and decision-making processes.”

Line 238-247:

“How may brain states modulate pupil-LC coupling? Pupil size changes have been linked to activity in other brain areas and neuromodulatory systems, including the medial prefrontal cortex, the inferior colliculus and cholinergic signaling (Joshi et al., 2016; Reimer et al., 2016; Okun et al., 2019; Kucyi and Parvizi, 2020; Pais-Roldán et al., 2020; Sobczak et al., 2021). A recent study found that pupil responses to dorsal raphe stimulation exhibited task uncertainty dependent variations (Cazettes et al., 2020). Therefore, it is possible that in high motivation/engagement states, multiple circuits including the LC synergistically influence pupil size changes, resulting in the apparently stronger pupil-LC coupling. Future experiments are needed to elucidate how pupil and LC interact with these brain circuits during different behavioral contexts and cognitive processes.”

Figure 1:1) During the first read, I had assumed that Figure 1 showed recordings from sessions where mice were awake but passive. After reaching Figure 4, I then became unclear about whether mice were performing the go/no-go task already in Figure 1.

We apologize for this confusion. We now state it clearly that all recordings in Figure 1, 2, 5 were acquired during behavior.

2) It is not specified whether the peak pupil size occurs at a consistent latency after the first cluster spike or whether the latency is cluster size dependent.

We believe this is related to comment #2. Our previous work shows that the latency between peak pupil dilation and LC spikes is around 2-3 s (Yang et al., 2021). Using a 6-s or 3-s window to identify peak pupil responses did not affect the results (Figure S2c), and the results in Figure 5 were not affected using the 3-s window (not shown).

3) A different way to assess the correlation between pupil and LC activity, as well as the latency of the pupil response, is to first compute a smooth instantaneous firing rate signal (convolved with some gaussian kernels for instance), and then perform a cross-correlation between the firing rate and the pupil signal. This analysis will directly reveal the strength of the correlation and lag between the two vectors. Using the instantaneous firing rate could be informative especially with regards to the results in Figure S2.

The suggested analysis has been done in our previous work (e.g., Figure 2b, Yang et al., 2021), and we showed that the latency between peak pupil dilation and LC spikes is around 2-3 s. We believe our current analysis has some advantages both conceptually and methodologically, including (1) the ability to identify distinct LC spiking patterns (clusters), which allows us (2) to quantitatively link the number of LC spikes to pupil diameter, and (3) to use pupil size changes to predict LC activity on a moment-by-moment basis. In addition, Figure S2 has been removed due to weak statistics and inconclusive evidence.

4) Related to comment 1 of the public review: here the authors may want to condition their analysis to epochs when mice are licking, whisking, etc. (assuming that these data have been recorded from the behavioral task). If there are also some video recordings of the face (not just the pupil), the authors can even track facial movements and correlate such signals with the pupil response and LC activity.

We believe this is similar to comment #1. Please see our response above. In brief, the new analysis showed that licking cannot account for the variability (Figure 5b). In addition, we performed the suggested analysis conditioned on licking periods only (± 0.5 s from each licking event). Fewer recordings exhibited monotonic relationships, likely due to fewer LC clusters during licking periods. However, the overall trend in Figure 5 still held (data not shown). We can include this result if deemed critical.

The reviewer raised a great point on how different types of movements may affect pupil-LC coupling, or in a more general sense, affect physiological and cognitive processes. We believe these are emerging and important questions and require separate and comprehensive investigations. We include a paragraph to discuss this topic, line 249-254:

“Another possibility is that higher engagement states may be intimately associated with more ‘uninstructed’ movements as revealed by recent work (Musall et al., 2019), which can drive robust neuronal activity throughout the brain (Musall et al., 2019; Steinmetz et al., 2019; Stringer et al., 2019; Salk_off_ et al., 2020). Future studies with comprehensive movement monitoring will determine whether more frequent movements, both task-related and task-unrelated, during periods of high motivation/engagement underlie the stronger pupil-LC coupling.”

Figure 3:1) The authors claim that they have recorded the same unit across different days. Yet, it is not described in the text nor in the methods what criteria are used to define those units as identical.

We have provided new data to support that the putatively same units were tracked, based on opto-tagging, spike clustering and waveform comparison (Figure 3c-g).

2) I appreciate that examples of single mice are shown (Figure 3, Figure S8-10), but it would also help to see a summary plot with all the mice, all the sessions, and all the different conditions.

We have included new data to show group pupil responses under different conditions (Figure 4c-h).

3) Related to the point above: assuming that there are significant session-to-session fluctuations in the pupil response to identical LC stimulation, could you describe it further? Does the stimulation of LC always induce a smaller pupil response on the second day? Is it consistent across mice?

Pupil responses in an earlier session were not consistently higher or lower than in a later session. This is now discussed in line 144-149:

“Group data from multiple mice further demonstrated that significant session-to-session fluctuations of pupil responses were prevalent but not directional (solid lines in Figure 4c, d), i.e., pupil responses in an earlier session (session 1) were not consistently higher or lower than in a later session (session 2). Therefore, such session-to-session fluctuations were not observable from group comparisons (Figure S8, Privitera et al., 2020).”

4) The 95% confidence intervals seem extremely small compared to their respective P-values.

This is now described in more details in Methods.

Figure 4:1) This manuscript is formatted as a Research Advances and builds on Yang et al., 2021. Yet, it would help to have a sentence or two to briefly describe the task, either here or in the methods.

We apologize for this oversight. Description of the behavioral task is now included in Methods.

2) It is my understanding that the data shown here are from the same units as the data Figure 1e. However, it is unclear whether the results come from the same sessions. If they do, why are the slopes in Figure 4 not the same as the slope in Figure 1e (right)?

Indeed, the data are in Figure 1e and old Figure 4 (new Figure 5). For the subset of 9 recordings included in Figure 5 (with > 100 trials and R^2^ > 0.6), their slopes shown in the y-axis of Figure 5a and in Figure 1e are the same.

3) The more impulsive the animal (negative bias), the stronger the coupling between pupil and LC. Therefore, here again, we cannot rule out the implication of movements, since impulsive mice may tend to respond stronger and faster (larger false alarm, larger hit rate, which also correlates with stronger pupil-LC coupling). Without this basic control for movement, it is difficult to believe that the LC-pupil relationship directly depends on cognitive states such as motivation and engagement as stated in the paper, rather than difference in movements.

We believe this is similar to comment #1 and recommendation #4. Please see our response above. In brief, we think these are emerging and important questions in the field: What is the relationship between brain states and movements? How could different types of movements affect physiological processes including the pupil-LC relationship studied here? We performed new analysis to show that one major type of movement in this task, licking, cannot account for the reported variability (Figure 5b). However, the reviewer is correct that other task-related or unrelated movements may come into play. We included a new section to discuss this topic, line 249-254:

“Another possibility is that higher engagement states may be intimately associated with more ‘uninstructed’ movements as revealed by recent work (Musall et al., 2019), which can drive robust neuronal activity throughout the brain (Musall et al., 2019; Steinmetz et al., 2019; Stringer et al., 2019; Salk_off_ et al., 2020). Future studies with comprehensive movement monitoring will determine whether more frequent movements, both task-related and task-unrelated, during periods of high motivation/engagement underlie the stronger pupil-LC coupling.”

[Editors’ note: what follows is the authors’ response to the second round of review.]

The new analyses and more tempered conclusions make the study more convincing compared to the previous version. However, there are remaining methodological and analysis concerns that should be addressed. If these concerns are addressed, there is agreement that the revised study would contribute to the field of LC physiology and how LC activity relates to pupil size, and would be suitable for publication in eLife.Please address all the concerns outlined by the reviewers below.In addition, it seems like it would be possible to perform an analysis of variability of optostim-to-dilation *within* each session and show that this variability is similar to the variability across sessions. This would help ameliorate concerns about variability in the LC-pupil relationship being due to variability in the long term stability of recordings and the optical fiber across sessions. The authors should provide information about the within-session vs cross-session variability.

We thank the editors for this suggestion. We performed the analysis where across-session variability was estimated by resampling trials pooled from all sessions and found that within session and across-session variability were comparable (new Figure 4—figure supplement 3). Furthermore, for the paired sessions that exhibited significantly different responses to optical stimulation, we tested whether across-session variability was comparable to within-session variability by resampling trials pooled from both sessions to generate a distribution of across session variability and examined whether the variability of individual sessions fell outside 5% of the distribution. We found that 2 session pairs out of 12 under awake 10-ms stimulation condition had larger across-session variability. 3 session pairs out of 11 under awake 50-ms stimulation condition had larger across-session variability. Altogether, we conclude that across-session and within-session variability were largely comparable. This is now included in the revised manuscript, e.g., line 160-164:

“Further analysis showed that across-session variability of pupil responses was comparable to within-session variability (Figure 4—figure supplement 3, Methods). In addition, for the paired sessions that exhibited significantly different responses to optical stimulation (solid lines in Figure 4c-f), only a small subset exhibited larger across-session variability than within-session variability (2 pairs out of 12 under 10-ms condition, and 3 pairs out of 11 under 50-ms condition, Methods).”

Reviewer #1:The authors main claim is that pupil diameter is not an accurate real-time readout of locus coeruleus activity. With the revised text, figures, and new analyses, the claim is convincing. I think the title change more accurately conveys the findings of the study.With the addition of new analyses, the authors convincingly show that there is substantial session-to-session variability in the pupil dilation response to optogenetic stimulation (Figure 4C, D, etc.). In response to the prior review, the authors addressed the differences in relation to Privitera, et al. (2020) regarding pupil response variability across sessions. Moreover, with the addition of a new analysis of waveform correlations for LC single units across days (Figure 3G) the authors convincingly show that the same units can be tracked across days. The authors then point out that, in contrast to the pupil response to opto-stim having variable responses across days, the response of the same unit to opto-stim on different days is not variable (Figure 3H). Based on these data, the authors suggest that LC neurons are not an "accurate real-time readout of locus coeruleus activity". I think these new analyses and the data collectively do support this statement.

We thank the reviewer for the positive evaluation.

There is one caveat to this, which is that different size populations of LC neurons could be driven by optical stimulation on different days. This could occur because slow changes in the tissue over days (gliosis, blood flow, inflammation) could occlude light or alter light transmission. Under this scenario, the neurons tracked across days were likely not affected by changes in inflammation/gliosis (which is why they could be tracked), but this does not mean that LC neurons which could not be monitored by the electrode did not receive different light transmission on different days. If different size populations were stimulated, then the overall pupil dilation might change. This caveat is of course nothing that the authors could address given the current technology limitations and it is mentioned just for their consideration and could be discussed if they agree and find this warranted.

We agree with this potential caveat. We have expanded the relevant Discussion section to incorporate this caveat more explicitly, line 224-227:

“The fact that the putatively same neurons tracked across days exhibited similar responses to optical stimulation cannot fully warrant the long-term stability of population LC response because slow changes in the tissue due to tetrode/optical fiber implant (gliosis, inflammation, etc.) could alter light transmission to the neurons that were not recorded.”

In addition, based on the editors’ comment above, we performed new analyses to show that within-session and across-session variability of pupil responses were comparable (new Figure 4figure supplement 3 and related discussion), to ameliorate concerns about long-term stability of LC responses.

There is a question about the number of units in Figure 3H. The legend states 5 awake units (3H, left panel) and does not report the number of units for anesthetized (3H, right panel). However, it appears that there are only 4 awake units (dots) in the left panel.

We apologize for this confusion. All 5 units were tested under the anesthetized condition (3 subsets tested with 3 different patterns of optical stimulation), and 4 out of the 5 units were tested in the awake condition (1 stimulation pattern). We now make this clear in the legends of Figure 3h.

There remains one methodological concern regarding single unit recordings.The authors added additional quantification of spike train and spike clustering characteristics. The new Figure S1A shows that the inter-spike interval (ISI) histogram has most of its density around a mean of 20-25 msec. However, the figure legend reports a median of 200 msec. That median does not appear to match the distribution plotted, given that nearly the entire distribution plotted in Figure S1A is <0.2 sec. In prior work that recorded awake monkey LC single units, the ISI histogram has been shown to peak between ~150 msec and 350 msec with ISI's <100 msec rarely being observed (Aston-Jones, G., Rajkowski, J., Kubiak, P. and Alexinsky, T. Locus coeruleus neurons in monkey are selectively activated by attended cues in a vigilance task. J Neurosci 14, 4467-4480 (1994)). Other studies of multi-unit activity have found a median ISI of 53 msec with an inter-quartile range of 20 to 157 msec (Kalwani, R. M., Joshi, S. and Gold, J. I. Phasic activation of individual neurons in the locus ceruleus/subceruleus complex of monkeys reflects rewarded decisions to go but not stop. J Neurosci 34, 13656 13669 (2014)). This range appears to be in line with the authors' presentation of the ISI distribution in Figure S1A. On the other hand, the example waveforms look convincing. Further quantification is necessary, such as showing some ISI histograms for individual single units as well as some waveform clustering examples and average waveforms. Given the small number of units in the study, I think it would be helpful for the field to see for each unit the average waveform and the ISI histogram for each unit in a supplementary figure. This can help the field to better interpret their work in the context of others. Additionally, a presentation of the distribution of waveform durations against firing rate would also provide helpful support for interpreting the results.

In Figure 1—figure supplement 1a (original Figure S1a), ISI was linearly binned and logarithmically plotted. We apologize that this plot did not clearly reveal the right side of the distribution. It is now changed to logarithmic binning for better visualization. This unit indeed has a median ISI of 200 ms (first example in the new Figure 1—figure supplement 1a). We now include additional examples of ISI distribution, spike waveform and spike clustering in Figure 1—figure supplement 1a, and list ISI and spike waveform from all recordings in the new Figure 4—figure supplement 6.

While the authors' claim that pupil diameter is not an accurate real-time readout of locus coeruleus activity is supported by their data, I am still not convinced that this is not already a known and accepted idea. The field clearly knows that many different neuronal circuits affect pupil size. For example, it is known that LC output (axonal activity) tracks specific aspects of pupil dynamics but cholinergic neurons another aspect (Reimer, J. et al. Pupil fluctuations track rapid changes in adrenergic and cholinergic activity in cortex. Nat Commun 7, 13289 (2016)). With that being said, I think these data support their claim and add to the literature on this topic.

We thank the reviewer for this comment. We hope our current manuscript will reinforce this understanding and contribute to the field.

Reviewer #3:The authors have performed new analyses (Figure 3, 4, S1, S2c, S8, S9) in an attempt to address most of the reviewers' comments. Missing information in the text and methods has been added, especially regarding the optogenetics experiments, and the discussion has been extended. This helps clarify some aspects of the paper. Some of the results are now more convincing (especially Figure 3 and 4) but other analyses remain oversimplified and therefore do not strongly support several conclusions.1) In Figure 1 and Figure S2, depending on the method used to quantify the relationship between pupil dilation and cluster size, a different number of neurons meet the R2>0.6 criterion. This seems problematic since following analyses only focus on this subset of neurons. Are there always the same neurons that consistently fall below the 0.6 criterion? Do the results hold when all the recorded neurons are included?

When the pupil-LC relationship was quantified using % changes from baseline (original Figure S2a, new Figure 1—figure supplement 3a), the exact same 13 recordings in Figure 1e were identified with R^2^ > 0.6. When the relationship was quantified using a 3-s window (new Figure 1—figure supplement 3c) instead of 6-s, 12 out of the 13 recordings in Figure 1e were identified with R^2^ > 0.6. Using the time derivative of pupil yielded a slightly lower number of linear pupil-LC relationship (new Figure 1figure supplement 3b, 9 recordings with R^2^ > 0.6), and 8 out of the 9 were the same as in Figure 1e. We expanded the legends of new Figure 1—figure supplement 3 to include this information. Furthermore, the results in Figure 1g held when all recordings were included. This is now shown in the new Figure 1—figure supplement 4.

2) I am still not fully convinced about the validity of the pupil dilation vs. cluster size analysis. Yet, this result (and its variability) is at the essence of the main conclusion of the paper. In particular, this analysis assumes that discrete events (spike clusters) linearly increase pupil size after a fixed latency (2-3s from Yang et al., 2021). Not accounting for the number of spikes occurring after a given cluster in the 3 s or 6 s time window disregard the fact that the latency may be variable or that the relationship between LC spiking and changes in pupil size could be nonlinear. This could reflect true variability in the LC-pupil relationship.

We apologize for not being clear in our previous response to a similar question. In addition to our recent work showing that the peak correlation coefficient between pupil and LC spiking has a lag of 2-3 s (Yang et al., 2021), other studies in rodents have demonstrated that activating the LC induced peak pupil dilation with a comparable latency (e.g., Breton-Provencher and Sur, 2018; Liu et al., 2017 and our own data in Figure 4). In our opinion, these lines of evidence strongly suggest that the pupil responds to LC activity with a temporal delay on the order of 2-3 s. Furthermore, based on the reviewer’s comment, we performed new analyses to show that the latency of peak pupil diameter following LC clusters did not significantly vary with cluster size (correlation coefficient = 0.37, P = 0.37), and ranged between 2.5 and 4 s (new Figure 1—figure supplement 2). We also found that only in 1 out of the 13 recordings (with R^2^ > 0.6) did the number of spikes occurring after a given cluster (in-between spikes) significantly correlate with LC cluster size or peak pupil diameter (P = 0.049 and 0.047, respectively), and overall the in-between spikes did not significantly correlate with LC cluster size (correlation coefficient = 0.35, P = 0.39; R^2^ = 0.10) or peak pupil diameter (correlation coefficient = 0.29, P = 0.48; R^2^ = 0.15). In our opinion, altogether these results strongly suggest that the in-between spikes did not significantly contribute to the variability of the pupil-LC relationship. This information is now included in the legends of Figure 1—figure supplement 3.

3) In Figure 5b, the non-liking period is defined as the activity outside a one second window around each licking events (clusters occurring within {plus minus}0.5 s from a lick were excluded). This choice is somewhat arbitrary, or at least not well motivated in the text. Besides, the pupil response is slow and the authors mentioned that the latency between peak pupil dilation and LC spikes is around 2 to 3 s. Therefore, the 1 s window may not be the most appropriate. A more convincing approach would be to condition the analysis on the number of licks performed in a trial for instance.

Our choice of the ± 0.5 s window was motivated by previous results that peak LC response occurred within a few hundred milliseconds of licking onset (Yang et al., 2021). This is now included in Methods. In addition, we did further analyses to exclude pupil responses in a 4-s window following each licking event. Fewer recordings passed the inclusion criterion (6 instead of 9), but the overall relationship held (Left panel in Author response image 1). We also conditioned the analysis on no-licking trials only, as suggested by the reviewer, and similar results held (Right panel in Author response image 1).

**Author response image 1. sa2fig1:**